# stUAI: Uncertainty-Aware Clustering of Spatially Resolved Transcriptomics Data

## Abstract

Spatially resolved transcriptomics (SRT) technologies enable high-resolution investigation of gene expression patterns across tissue sections, providing unprecedented insights into the molecular architecture of tissues. However, the inherent sparsity and over-dispersion of gene expression across spots, together with view-specific heterogeneity, jointly complicate modeling and present formidable challenges to reliable spatially informed downstream analyses. To address these issues, we propose stUAI, an Uncertainty-Aware Integration framework for spatially transcriptomics data. stUAI first learns spatial- and expression-view embeddings by two separate graph-based encoders to capture multi-scale information. Then stUAI exploits the distributional representation of spots to quantify the view-specific uncertainty caused by technology limitations or noise in SRT, which is further leveraged for intra-view contrastive learning, cross-view information alignment, and representation integration. Furthermore, stUAI incorporates a zero-inflated negative binomial decoder to handle expression sparsity and imposes spatial structural constraints to preserve spatial continuity. Extensive experimental results on multiple benchmark datasets validate the effectiveness of our proposed stUAI in spatial clustering and several downstream applications.

## 1 Introduction

Recent advances in spatially resolved transcriptomics (SRT) have enabled the simultaneous measurement of gene expression and spatial localization, offering a new perspective on tissue architecture and cellular organization (Rao et al., 2021). While single-cell RNA sequencing (scRNA-seq) has significantly advanced our understanding of cellular heterogeneity (Xu et al., 2023; Erhard et al., 2022), it inherently lacks spatial context, which is crucial for deciphering the functional roles of cells within their native microenvironments. As a result, *spatial clustering*, which aims to uncover region-specific cell populations by integrating gene expression and spatial information (Williams et al., 2022), has emerged as a key task in SRT analysis.

SRT technologies can be broadly classified into two categories: imaging-based in situ methods, such as MERFISH (Moffitt et al., 2018), osmFISH (Codeluppi et al., 2018), and barcode-based sequencing approaches, including 10x Visium (Ji et al., 2020), and Stereo-seq (Chen et al., 2022). Although these platforms have gained widespread adoption, they often yield data with high noise levels and substantial dropout events (Wang et al., 2023), which pose significant challenges for downstream analyses. Traditional clustering algorithms, such as $k$-means (Likas et al., 2003) and Louvain (Blondel et al., 2008), typically overlook spatial dependencies and thus struggle to capture the complex spatial organization of tissues. To overcome these limitations, graph neural networks (GNNs) (Kipf & Welling, 2017) have emerged as powerful tools for spatial clustering due to their ability to model non-Euclidean structures. For instance, SpaGCN (Hu et al., 2021) and DeepST (Xu et al., 2022) integrate multimodal signals, including gene expression, spatial coordinates, and histological features, within a graph-based learning framework. Other approaches, such as GraphST (Long et al., 2023), perform joint representation learning to effectively fuse transcriptomic and spatial modalities. More recently, DUSTED (Zhu et al., 2025) combines graph autoencoding with attention mechanisms and negative binomial modeling to capture spatial features and gene expression noise, and Spotscape (Oh et al., 2025) leverages global relationship modeling and a similarity scaling strategy to achieve effective multi-slice integration beyond local adjacency. Overall, these methods effectively capture spatial patterns in SRT data for more accurate and biological spatial domain detection.

While a variety of spatial clustering methods have been developed from different perspectives with promising results, they still exhibit intrinsic limitations. ***In particular***, although most approaches emphasize high noise levels in SRT technologies, resulting in high sparsity and over-dispersion of the data, many of them address noise only implicitly, e.g., some autoencoder-based methods (Long et al., 2023; Xu et al., 2024). As a result, these methods may struggle to denoise effectively and leave model uncertainty, which reflects the inherent semantic ambiguity. ***Moreover***, contrastive learning, e.g., deep graph informax-based methods (Liu et al., 2023; Long et al., 2023), has been widely adopted to obtain discriminative latent representations, yet reliable guidance for constructing meaningful positive and negative pairs is often lacking. This often leads to erroneous or overconfident discriminative power. ***Finally***, while numerous methods attempt to integrate complementary information, i.e., spatial and expression modality, through representation fusion (Zhu et al., 2024; Zhang et al., 2025), they typically lack dynamic mutual guidance and knowledge sharing between modality-specific representations, which impedes the semantic richness of the integrated representations. These limitations ultimately hinder the performance of spatial clustering and also subsequent downstream tasks. Therefore, *it is crucial to develop more effective noise-handling strategies that incorporate uncertainty explicitly and improve knowledge sharing across modalities.*

Building upon these insights, we develop the **U**ncertainty-**A**ware **I**ntegration framework for **s**patially resolved **t**ranscriptomics data clustering (stUAI). stUAI first constructs spatial- and feature-level graphs based on spatial positions and gene expression profiles, and then applies separate encoders to form a two-channel architecture. Based on the two views, stUAI quantifies uncertainty of both topology and feature by modeling distributional representations for spots (Vilnis & McCallum, 2015; Alejandro et al., 2023). These distributional distributions are then incorporated into model optimization. Specifically, to learn discriminative latent representations, stUAI first performs intra-view contrastive learning. The augmented samples are generated by sampling from the distributional representations, while contrastive pairs are constructed under the guidance of high-confidence clustering assignments, thereby improving the stability and reliability of discriminative representations. To enable mutual guidance and knowledge transfer between views, stUAI perform cross-view optimal transport based on their respective distributional representations, achieving distributional alignment and encouraging semantically enriched latent representations. Finally, stUAI incorporates a zero-inflated negative binomial (ZINB) decoder for feature reconstruction, which further facilitates denoising on the sparse SRT data, and imposes structural constraints on the latent representations to reinforce spatial supervision. Extensive experiments on benchmark SRT datasets demonstrate that stUAI surpasses state-of-the-art methods, confirming its superiority in spatial clustering.

## 2 METHODOLOGY

**Preliminaries.** Spatial clustering for SRT data aims to integrate spatial positions and gene expression profiles to obtain informative and discriminative latent representations, used for spatial domain identification and cellular heterogeneity analysis. Let $\mathbf{S} = \{(x_i, y_i)\}_{i=1}^{N}$ denote the set of spatial coordinates associated with $N$ spots; $\mathbf{F} \in \mathbb{R}^{N \times D}$ denote the gene expression matrix, where $D$ is the number of genes (after pre-processing). Each row $\mathbf{f}_i \in \mathbb{R}^D$ of $\mathbf{F}$ represents the gene expression profile at spot $i$ in the tissue. The goal is to learn a latent representation $\mathbf{h} \in \mathbb{R}^{N \times d}$ with $d \ll D$, which effectively integrates both gene expression $\mathbf{F}$ and spatial coordinates $\mathbf{S}$, such that $\mathbf{h} = f(\mathbf{F}, \mathbf{S})$, where $f(\cdot)$ is a learning function that captures both cellular and spatial structures. The learned representation $\mathbf{h}$ is then used to perform spatial clustering, and $\{\mathcal{C}_1, \ldots, \mathcal{C}_K\}$ denotes the resulting cluster assignments, corresponding to spatial domains within the tissue.

### 2.1 PROXIMITY GRAPH CONSTRUCTION AND UNCERTAINTY QUANTIFICATION

SRT offers the advantage of leveraging spatial pattern to identify similar cells. To fully exploit spatial information and harness the powerful abilities of GNNs, we first convert the spatial coordinates and expression profiles into a spatial-level undirected graph $\mathcal{G}^{(s)} = (\mathbf{A}^{(s)}, \mathbf{F})$, where adjacency matrix $\mathbf{A}^{(s)}$ is constructed using fixed-radius nearest neighbors, i.e., $\mathbf{A}_{ij}^{(s)} = 1$ if the Euclidean distance between spots $i$ and $j$ is less than a predefined radius $r$, and $\mathbf{A}_{ij}^{(s)} = 0$ otherwise. In complex tissues, cells of the same type may not be spatially adjacent. To account for this, we additionally construct a feature-level undirected graph based on gene expression matrix. This graph $\mathcal{G}^{(f)} = (\mathbf{A}^{(f)}, \mathbf{F})$, is

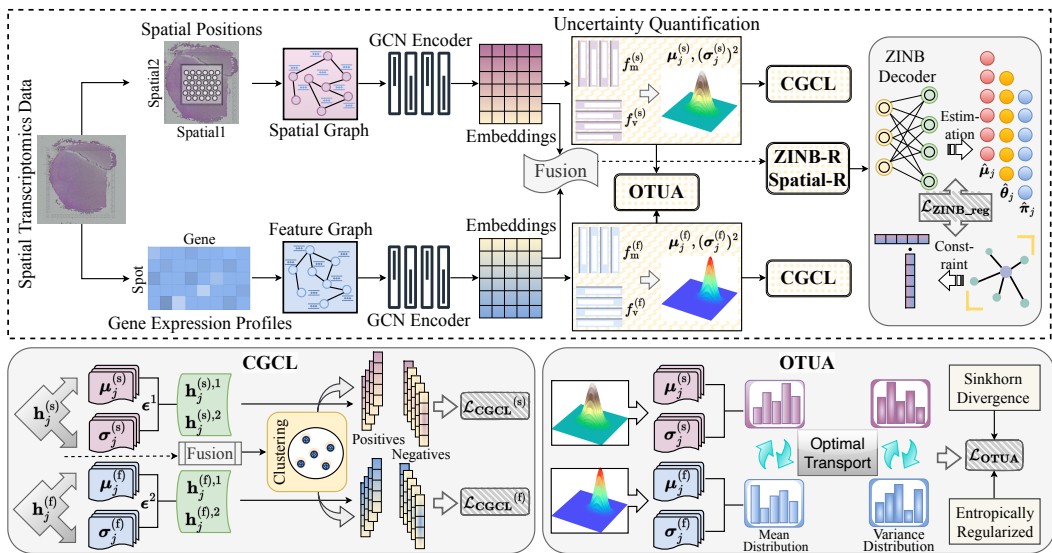

Figure 1: Illustration of the proposed framework stUAI.

constructed using $k$-nearest neighbors. Here, $\mathbf{A}_{ij}^{(\mathrm{f})} = 1$ indicates that spots $i$ and $j$ are among each other's top-$k$ nearest neighbors (measured by cosine similarity); otherwise, $\mathbf{A}_{ij}^{(\mathrm{f})} = 0$.

In SRT analysis, graph convolution network (GCN, Kipf & Welling, 2017) plays a crucial role in capturing the complex relationships between topology locations and gene expression profiles (Hu et al., 2021; Long et al., 2023; Zhang et al., 2024). To effectively integrate both spatial- and feature-level graphs, $\mathcal{G}^{(\mathrm{s})}$ and $\mathcal{G}^{(\mathrm{f})}$, we utilize two separate GCNs to encode them independently. Formally, the layer-wise propagation rule for a GCN is defined as:

$$\mathbf{H}^{(i,l+1)} = \sigma(\hat{\mathbf{A}}^{(i)}\mathbf{H}^{(i,l)}\mathbf{W}^{(i,l)}), \ i \in \{\mathrm{s},\mathrm{f}\}$$

where $\mathbf{H}^{(i,l)}$ is the node embedding matrix at layer $l$ (with $\mathbf{H}^{(i,0)} = \mathbf{F}$), $\mathbf{W}^{(i,l)}$ is the learnable weight matrix, $\sigma(\cdot)$ is a nonlinear activation function (e.g., ReLU), and $\hat{\mathbf{A}}^{(i)}$ is the normalized adjacency matrix with self-loops, computed as $\hat{\mathbf{A}}^{(i)} = (\tilde{\mathbf{D}}^{(i)})^{-\frac{1}{2}}\tilde{\mathbf{A}}^{(i)}(\tilde{\mathbf{D}}^{(i)})^{-\frac{1}{2}}$, where $\tilde{\mathbf{A}}^{(i)} = \mathbf{A}^{(i)} + \mathbf{I}$ and $\tilde{\mathbf{D}}^{(i)}$ is the degree matrix of $\tilde{\mathbf{A}}^{(i)}$. The outputs of the two GCNs are denoted as $\mathbf{H}^{(\mathrm{s})} = (\mathbf{h}_1^{(\mathrm{s})}, \ldots, \mathbf{h}_N^{(\mathrm{s})})^\top$ and $\mathbf{H}^{(\mathrm{f})} = (\mathbf{h}_1^{(\mathrm{f})}, \ldots, \mathbf{h}_N^{(\mathrm{f})})^\top$, respectively.

Current sequencing technologies often introduce a high level of noise (Long et al., 2023), which persists even after pre-processing. Moreover, the above graph construction relies on hard thresholds and produces binary adjacency matrices, inevitably resulting in information loss. These factors jointly contribute to the uncertainty in the learned node embeddings and the subsequent model predictions. However, single-point embeddings is insufficient to capture uncertainty. A prominent approach to modeling uncertainty in the embedding space is to employ Gaussian distributions (Vilnis & McCallum, 2015; Yang et al., 2021). Hence, we build upon above learned node embeddings and leverage Gaussian distributions to learn stochastic distributional representations, thereby characterizing the uncertainty induced by the inputs. Specifically, we define the distributional representation as:

$$\bar{\mathbf{h}}_j^{(i)} \mid \mathbf{h}_j^{(i)} \sim \mathcal{N}\left(\boldsymbol{\mu}_j^{(i)}, \mathrm{diag}((\boldsymbol{\sigma}_j^{(i)})^2)\right), \ \boldsymbol{\mu}_j^{(i)} = f_\mathrm{m}^{(i)}(\mathbf{h}_j^{(i)}), \ \log((\boldsymbol{\sigma}_j^{(i)})^2) = f_\mathrm{v}^{(i)}(\mathbf{h}_j^{(i)}), \quad (1)$$

where $\mathcal{N}(\cdot, \cdot)$ denotes the Gaussian distribution, $f_\mathrm{m}^{(i)}$ and $f_\mathrm{v}^{(i)}$ are fully connected mean and log variance networks for $j \in [1 : N], i \in \{\mathrm{s}, \mathrm{f}\}$. The distributional representation provides a fuzzy depiction of graph semantics, underscoring spot ambiguity induced by artificial or technical noise. Unlike Bayesian or Monte Carlo methods (Kendall & Gal, 2017), our approach is computationally efficient. In next section, we incorporate the distributional representations into training and integration to promote stable, discriminative and semantically meaningful representations for spatial clustering, while iterative optimization further refines representation learning via the mean and variance networks. Then the learned variances can serve as effective measures of uncertainty, where high-quality spots have small variance, whereas low-quality spots exhibit large variance.

## 2.2 Cluster-Guided Intra-View Contrastive Learning

To learn discriminative distributional representations at both the spatial and feature levels, we first employ contrastive learning (You et al., 2020) to enhance representation learning. Traditional methods typically apply random perturbations to nodes or graph topology for augmentation, which are prone to produce prominent outlier expressions or noise in highly sparse and over-dispersed SRT data, causing the contrastive learning to be dominated by extreme values, thereby exacerbating the instability and semantic bias of the embeddings. Moreover, due to the lack of ground-truth labels in spatial clustering, the construction of positive and negative pairs is often unreliable. Using only self-augmentations as positives fails to leverage the shared patterns among similar cells, further reducing the robustness of embeddings to sparse and over-dispersed data. To address these issues, we perform augment-free contrastive learning by sampling from the distributional representations to mitigate semantic bias; and we construct reliable positive and negative pairs based on high-confidence clustering results. This encourages stability in representations of semantically similar spots and enhances the discriminability of representations across different semantic spots.

Specifically, we perform differentiable sampling via reparameterization trick (Kingma et al., 2013):

$$\mathbf{h}_j^{(i),1} = \boldsymbol{\mu}_j^{(i)} + \boldsymbol{\epsilon}^1 \odot \boldsymbol{\sigma}_j^{(i)}, \ \mathbf{h}_j^{(i),2} = \boldsymbol{\mu}_j^{(i)} + \boldsymbol{\epsilon}^2 \odot \boldsymbol{\sigma}_j^{(i)}, \ \boldsymbol{\epsilon}^1, \boldsymbol{\epsilon}^2 \sim \mathcal{N}(\mathbf{0}, \mathbf{I}_d), j \in [1:N], \ i \in \{\mathrm{s}, \mathrm{f}\}, \quad (2)$$

where $\odot$ denotes element-wise multiplication, and $d$ represents the dimensionality of the node embeddings. Therefore, $\mathbf{h}_j^{(i),1}$ and $\mathbf{h}_j^{(i),2}$ can be regarded as an augmented pair with similar semantics and are naturally treated as a positive pair. In addition, it is also reasonable to consider spots predicted to the same category as positive pairs. To this end, we utilize the high-confidence clustering results based on the fused representations (to be introduced in Section 2.4) from the previous training epoch to determine the positive pairs in the current epoch. Concretely, we set a confidence threshold $\kappa \in [0, 1]$ and compute a distance vector $\mathbf{d} \in \mathbb{R}^N$, each element of which represents the distance between a specific spot and its assigned cluster centroid. We then identify the top $(1 - \kappa)N$ nodes with the smallest values in $\mathbf{d}$ as high-confidence nodes. Let $\mathbf{m} \in \{0, 1\}^N$ denote a binary confidence mask, where $\mathbf{m}_j = 1$ if spot $j$ is selected as high-confidence and $\mathbf{y} \in \{1 : K\}^N$ be the predicted label vector for $K$ clusters. The binary positive pair matrix $\mathbf{R} \in \{0, 1\}^{N \times N}$ is then defined as $\mathbf{R} = (\mathbf{m}\mathbf{m}^\top) \wedge (\mathbf{y} = \mathbf{y}^\top)$, where the logical AND operation $\wedge$ ensures that a node pair is considered positive only if both nodes are high-confidence and share the same predicted label. Accordingly, the cluster-guided contrastive learning (CGCL) loss is defined as:

$$\mathcal{L}_{\mathrm{CGCL}} = \frac{1}{2}(\mathcal{L}_{\mathrm{CGCL}}^{(\mathrm{s})} + \mathcal{L}_{\mathrm{CGCL}}^{(\mathrm{f})}), \ \text{where} \ \mathcal{L}_{\mathrm{CGCL}}^{(i)} = -\frac{1}{2N} \sum_{j=1}^{N} \left[ h(\mathbf{h}_j^{(i),1}, \mathbf{H}^{(i),2}) + h(\mathbf{h}_j^{(i),2}, \mathbf{H}^{(i),1}) \right],$$

$$\text{and} \ h(\mathbf{h}_j^{(i),1}, \mathbf{H}^{(i),2}) = \log \frac{\sum_{j' \in \mathcal{I}_j} \exp(\mathrm{sim}(\mathbf{h}_j^{(i),1}, \mathbf{h}_{j'}^{(i),2})/\tau)}{\sum_{j'' \in [1:N] \setminus \mathcal{I}_j} \exp(\mathrm{sim}(\mathbf{h}_j^{(i),1}, \mathbf{h}_{j''}^{(i),2})/\tau)}, i \in \{\mathrm{s}, \mathrm{f}\}, \quad (3)$$

where $\mathcal{I}_j$ is the positive index set for spot $j$, initialized with $\{j\}$ and updated from the $j$-th row of $\mathbf{R}$, $\mathrm{sim}(\cdot, \cdot)$ is cosine similarity, and the temperature parameter $\tau$ is set to 0.5 (You et al., 2020).

## 2.3 Cross-View distributional Representation Alignment

Spatial- and feature-level distributional representations inherently carry distinct information. To facilitate semantic consistency across different views and mitigate view-specific noise, we enforce alignment between their distributional representations. Optimal transport (OT, Villani et al., 2008) provides a principled framework for measuring the distance between probability distributions by capturing the minimal cost required to morph one distribution into another. Compared with KL and JS divergences, OT better handles distributions with differing supports and aligns complex high-dimensional distributions. In addition, the distributional representations are Gaussian, which can be fully characterized by their first- and second-order moments. Therefore, we adopt OT to align the normalized means and variances of distributional representations across the complementary views.

Specifically, we first normalize the mean and variance vectors $\boldsymbol{\mu}_j^{(i)}, \log(\boldsymbol{\sigma}_j^{(i)})^2$ (e.g., via Softmax) to obtain probability distributions $\mathcal{M}_j^{(i)}, \mathcal{V}_j^{(i)}, i \in \{\mathrm{s}, \mathrm{f}\}$. The entropically regularized OT problem (Yu et al., 2024; Vo et al., 2024) with marginal constraints is then formulated as: ($\mathcal{P} \in \{\mathcal{M}, \mathcal{V}\}$)

$$\mathrm{OT}(\mathcal{P}_j^{(\mathrm{s})}, \mathcal{P}_j^{(\mathrm{f})}) = \min_{\mathbf{P}} \langle \mathbf{C}, \mathbf{P} \rangle_{\mathrm{F}} + \epsilon \sum_{k,q} \mathbf{P}_{kq} \log(\mathbf{P}_{kq}), \text{s.t.} \ \mathbf{P} \in \mathbb{R}_+^{d \times d}, \mathbf{P}\mathbf{1} = \mathcal{P}_j^{(\mathrm{s})}, \mathbf{P}^\top \mathbf{1} = \mathcal{P}_j^{(\mathrm{f})},$$

where $\mathbf{C}$ is the cost matrix (e.g., based on Euclidean distances), $\mathbf{P}$ is the transport plan, and $\langle \cdot, \cdot \rangle_{\mathrm{F}}$ denotes the Frobenius dot-product. Then we utilize the Sinkhorn divergence to form the OT-based uncertainty alignment (OTUA) loss as:

$$\mathcal{L}_{\mathrm{OTUA}} = \frac{1}{2}\left( \mathcal{L}_{\mathrm{OTUA}}^{\mathcal{M}^{(s)},\mathcal{M}^{(f)}} + \mathcal{L}_{\mathrm{OTUA}}^{\mathcal{V}^{(s)},\mathcal{V}^{(f)}} \right), \tag{4}$$

with $\mathcal{L}_{\mathrm{OTUA}}^{\mathcal{P}^{(s)},\mathcal{P}^{(f)}} = \sum_{j=1}^{N} \left[ \mathrm{OT}(\mathcal{P}_j^{(s)}, \mathcal{P}_j^{(f)}) - \frac{1}{2}\left( \mathrm{OT}(\mathcal{P}_j^{(s)}, \mathcal{P}_j^{(s)}) + \mathrm{OT}(\mathcal{P}_j^{(f)}, \mathcal{P}_j^{(f)}) \right) \right], \mathcal{P} \in \{\mathcal{M}, \mathcal{V}\}$.

## 2.4 Representation Fusion and ZINB-Based Reconstruction

**Robust Representation Fusion**. To better support spatial clustering, we fuse the latent representations learned from the spatial and feature views to obtain semantically enriched spot representations. Specifically, we assign weights to each view based on the inverse variances from the distributional representations, so that the representation component with higher uncertainty contributes less to the fused representation. It ensures that the integrated representation retains more information. Mathematically, the fusion is performed as follows:

$$\mathbf{H} = \mathbf{\Omega}^{(s)} \odot \mathbf{H}^{(s)} + \mathbf{\Omega}^{(f)} \odot \mathbf{H}^{(f)}, [\mathbf{\Omega}^{(s)}, \mathbf{\Omega}^{(f)}] = \mathrm{Softmax}(\mathbf{\Sigma}^{(s)}, \mathbf{\Sigma}^{(f)}), \tag{5}$$

where $\mathbf{H} = (\mathbf{h}_1, \ldots, \mathbf{h}_N)^{\top}$, $\mathbf{\Sigma}^{(i)} = (1/(\boldsymbol{\sigma}_1^{(i)})^2, \ldots, 1/(\boldsymbol{\sigma}_N^{(i)})^2)^{\top}$ for $i \in \{\mathrm{s}, \mathrm{f}\}$, and $\mathrm{Softmax}$ is performed element-wise. In each training epoch, representation fusion and $k$-means clustering are performed to update the positive pair matrix $\mathbf{R}$ and also the contrastive loss $\mathcal{L}_{\mathrm{CGCL}}$ in Equation 3.

**ZINB-based Reconstruction and Spatial Regularization**. In addition, to better handle the high sparsity and over-dispersion inherent in transcriptomics data, we adopt a Zero-Inflated Negative Binomial (ZINB, Yu et al., 2022) model-based decoder to reconstruct the original expression patterns. It enables effective denoising and facilitates the discovery of the underlying data distribution. With $\mathrm{NB}(\mathbf{f} \mid \boldsymbol{\mu}, \boldsymbol{\theta}) = \Gamma(\mathbf{f} + \boldsymbol{\theta}) [\boldsymbol{\theta}/(\boldsymbol{\theta} + \boldsymbol{\mu})]^{\boldsymbol{\theta}} [\boldsymbol{\mu}/(\boldsymbol{\theta} + \boldsymbol{\mu})]^{\mathbf{f}} / [\mathbf{f}! \Gamma(\boldsymbol{\theta})]$, the ZINB distribution is:

$$\mathrm{ZINB}(\mathbf{f} \mid \boldsymbol{\mu}, \boldsymbol{\theta}, \boldsymbol{\pi}) = \boldsymbol{\pi}\delta_0(\mathbf{f}) + (1 - \boldsymbol{\pi})\,\mathrm{NB}(\mathbf{f}),$$

where $\boldsymbol{\mu}$ captures the expected gene expression, $\boldsymbol{\theta}$ models the over-dispersion typical of count data, $\boldsymbol{\pi}$ accounts for the dropout events caused by technical noise or low expression levels in SRT data, and $\delta_0(\cdot)$ is a dropout indicator function. To estimate the ZINB parameters, we employ three fully connected networks that map each fused representation vector $\mathbf{h}_j$ to each parameter, enabling reconstruction of the original expression profile and effective denoising:

$$\hat{\boldsymbol{\mu}}_j = \exp(f_{\mathrm{D}}^1(\mathbf{h}_j)), \hat{\boldsymbol{\theta}}_j = \mathrm{Softplus}(f_{\mathrm{D}}^2(\mathbf{h}_j)), \hat{\boldsymbol{\pi}}_j = \mathrm{Sigmoid}(f_{\mathrm{D}}^3(\mathbf{h}_j)), j \in [1:N], \tag{6}$$

where $f_{\mathrm{D}}^1, f_{\mathrm{D}}^2, f_{\mathrm{D}}^3$ are decoder networks. With the estimators, the reconstruction loss is designed by the negative log-likelihood of the ZINB distribution:

$$\mathcal{L}_{\mathrm{ZINB}} = -\sum_{j=1}^{N} \log \mathrm{ZINB}(\mathbf{f}_j \mid \hat{\boldsymbol{\mu}}_j, \hat{\boldsymbol{\theta}}_j, \hat{\boldsymbol{\pi}}_j).$$

In addition, we impose a structural constraint in the latent space, aiming for the learned representations to preserve the original spatial patterns. To this end, we perform spatial regularization by:

$$\mathcal{L}_{\mathrm{reg}} = -\sum_{j=1}^{N} \left( \sum_{\mathbf{A}_{jk}^{(s)}=1} \log(\sigma(\mathrm{sim}(\mathbf{h}_j, \mathbf{h}_k))) + \sum_{\mathbf{A}_{jq}^{(s)}=0} \log(1 - \sigma(\mathrm{sim}(\mathbf{h}_j, \mathbf{h}_q))) \right),$$

where $\sigma(\cdot)$ is an activation function that maps the cosine similarity values to the range $[0, 1]$. The reconstruction loss and the spatial regularization jointly contribute to the following loss:

$$\mathcal{L}_{\mathrm{ZINB\_reg}} = \mathcal{L}_{\mathrm{ZINB}} + \mathcal{L}_{\mathrm{reg}}. \tag{7}$$

## 2.5 Joint Optimization

Building upon the preceding discussions, we propose a unified framework stUAI, which jointly optimizes three losses: the cluster-guided contrastive learning loss in Equation 3, the OT-based uncertainty alignment loss in Equation 4, and the ZINB-based reconstruction loss with spatial regularization in Equation 7. In summary, the total loss of stUAI is designed as:

$$\mathcal{L}_{\mathrm{Total}} = \mathcal{L}_{\mathrm{CGCL}} + \mathcal{L}_{\mathrm{OTUA}} + \mathcal{L}_{\mathrm{ZINB\_reg}}. \tag{8}$$

The overall optimization process of the proposed stUAI is summarized in Appendix A.

Table 1: The spatial clustering performance on DLPFC dataset. The best and the second-best results in all the methods are highlighted with red and yellow, respectively.

| Slice | Metric | SCANPY | SpaGCN | DeepST | STAGATE | GraphST | Smoother | SCGDL | stLearn | DUSTED | stUAI (Ours) |
|---|---|---|---|---|---|---|---|---|---|---|---|
| 151507 | ARI | 0.20 | 0.39 | 0.46 | 0.54 | 0.48 | 0.20 | 0.49 | 0.49 | 0.46 | **0.70** |
| | NMI | 0.21 | 0.49 | 0.64 | 0.66 | 0.64 | 0.38 | 0.55 | 0.64 | 0.65 | **0.74** |
| 151508 | ARI | 0.15 | 0.33 | 0.46 | 0.49 | 0.49 | 0.24 | 0.34 | 0.31 | 0.45 | **0.65** |
| | NMI | 0.21 | 0.43 | 0.61 | 0.63 | 0.54 | 0.37 | 0.44 | 0.53 | 0.64 | **0.70** |
| 151509 | ARI | 0.19 | 0.35 | 0.48 | 0.53 | 0.52 | 0.29 | 0.32 | 0.45 | 0.45 | **0.58** |
| | NMI | 0.27 | 0.51 | 0.62 | **0.66** | 0.64 | 0.38 | 0.48 | 0.62 | 0.64 | **0.66** |
| 151510 | ARI | 0.14 | 0.37 | 0.53 | 0.41 | 0.50 | 0.18 | 0.31 | 0.44 | 0.49 | **0.70** |
| | NMI | 0.22 | 0.50 | 0.63 | 0.61 | 0.64 | 0.31 | 0.45 | 0.59 | 0.62 | **0.72** |
| 151669 | ARI | 0.10 | 0.23 | 0.42 | 0.35 | 0.48 | 0.21 | 0.24 | 0.32 | 0.47 | **0.50** |
| | NMI | 0.16 | 0.36 | 0.53 | 0.58 | 0.59 | 0.35 | 0.38 | 0.49 | 0.59 | **0.60** |
| 151670 | ARI | 0.09 | 0.33 | 0.36 | 0.32 | 0.46 | 0.23 | 0.26 | 0.23 | 0.29 | **0.49** |
| | NMI | 0.16 | 0.43 | 0.54 | 0.53 | **0.68** | 0.39 | 0.36 | 0.41 | 0.51 | 0.61 |
| 151671 | ARI | 0.12 | 0.42 | 0.48 | 0.51 | 0.61 | 0.44 | 0.31 | 0.39 | 0.50 | **0.84** |
| | NMI | 0.24 | 0.53 | 0.63 | 0.65 | 0.72 | 0.51 | 0.41 | 0.54 | 0.65 | **0.78** |
| 151672 | ARI | 0.12 | 0.52 | 0.44 | 0.54 | 0.63 | 0.47 | 0.34 | 0.34 | 0.57 | **0.79** |
| | NMI | 0.23 | 0.60 | 0.59 | 0.66 | 0.61 | 0.53 | 0.46 | 0.47 | 0.67 | **0.76** |
| 151673 | ARI | 0.20 | 0.40 | 0.57 | 0.45 | **0.63** | 0.48 | 0.33 | 0.30 | 0.57 | 0.56 |
| | NMI | 0.29 | 0.55 | 0.71 | 0.63 | **0.74** | 0.58 | 0.42 | 0.49 | 0.70 | 0.66 |
| 151674 | ARI | 0.22 | 0.31 | 0.48 | 0.48 | 0.43 | 0.38 | 0.27 | 0.38 | 0.46 | **0.55** |
| | NMI | 0.31 | 0.46 | 0.60 | 0.58 | 0.61 | 0.46 | 0.38 | 0.54 | 0.63 | **0.65** |
| 151675 | ARI | 0.23 | 0.27 | 0.52 | 0.36 | 0.55 | 0.30 | 0.30 | 0.38 | 0.56 | **0.58** |
| | NMI | 0.32 | 0.41 | 0.65 | 0.50 | 0.62 | 0.43 | 0.41 | 0.56 | 0.65 | **0.67** |
| 151676 | ARI | 0.22 | 0.31 | 0.50 | 0.49 | **0.61** | 0.31 | 0.29 | 0.40 | 0.52 | 0.55 |
| | NMI | 0.31 | 0.48 | 0.60 | 0.63 | 0.66 | 0.46 | 0.42 | 0.56 | 0.67 | **0.68** |

## 3 EXPERIMENT

### 3.1 EXPERIMENTAL SETUP

We evaluate our proposed stUAI on three datasets for spatial clustering, including the LIBD study of the human dorsolateral prefrontal cortex (DLPFC, Maynard et al., 2021), human breast cancer samples (HBC, Buache et al., 2011) and mouse anterior brain tissue (MAB, Dong, 2008). And we compare our stUAI with various baselines, including *non-spatial clustering* method, i.e., SCANPY (Wolf et al., 2018), and *spatial clustering* methods, i.e., SpaGCN (Hu et al., 2021), STAGATE (Dong & Zhang, 2022a), GraphST (Long et al., 2023), Smoother (Su et al., 2023), SCGDL (Liu et al., 2023), stLearn (Pham et al., 2023), DUSTED (Zhu et al., 2025), and *image-enhanced clustering* methods, i.e., DeepST (Xu et al., 2022). In experiments, we adopt *supervised* metrics, i.e., ARI (Vinh et al., 2009), NMI (Pfitzner et al., 2009), and Jaccard similarity coefficient (Levandowsky & Winter, 1971), and *unsupervised* metrics, i.e., Moran's I score (Moran, 1950), to evaluate the performance of spatial clustering. Detailed metrics and implement details can be found in Appendix I.

### 3.2 EXPERIMENTAL RESULTS

To evaluate the performance of our stUAI, we compare several competitive baselines for spatial clustering, as shown in Table 1, Figure 2 and Figure 3. The results reveal that spatially-aware methods consistently outperform non-spatial ones, as spatial information enhances the model's ability to capture tissue heterogeneity. stUAI achieves best results across all three datasets and significantly surpasses the runner-up on many tissue slices. For example, on slice 151671, stUAI outperform the runner-up method by 37.70% and 8.33% in two metrics, highlighting the superiority of our approach. This advantage arises from our method's ability to effectively leverage the intrinsic uncertainty for multimodal fusion and enrich embed-

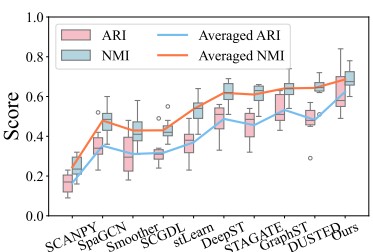

Figure 3: Boxplots of ARI and NMI on all DLPFC slices.

ding semantics by aligning spatial and feature distributions through OT. In Figure 2(a), we observe stUAI capturing the heterogeneity of HBC dataset by separating healthy tissue, DCIS/LCIS, IDC, and tumor edges, with gene expression patterns aligned with pathological features. In Figure 2(b), stUAI accurately delineates cortical layers and white matter in the DLPFC dataset, which signifies

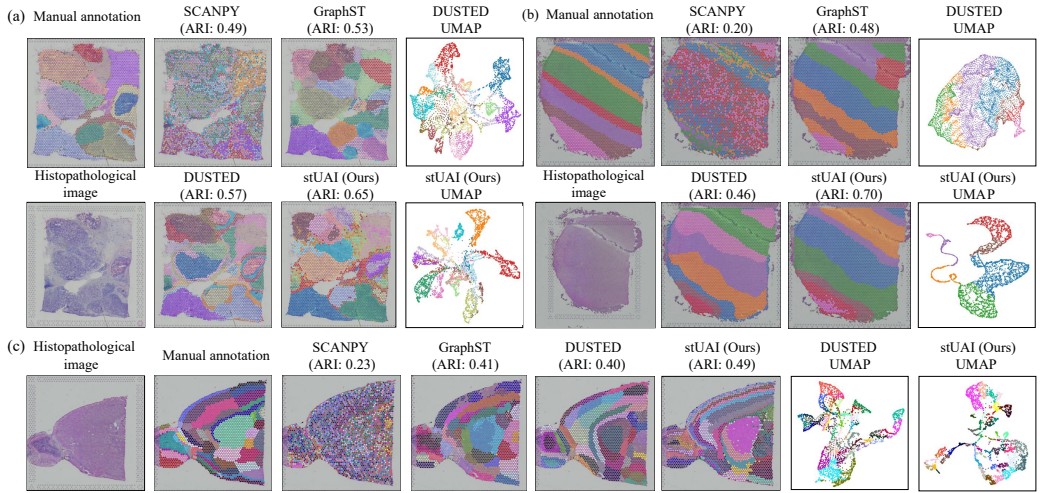

Figure 2: Spatial and UMAP visualizations of spatial clustering on (a) HBC, (b) DLPFC (151507), and (c) MAB tissue datasets.

strong clustering while preserving anatomical structure. In Figure 2(c), stUAI applies to the MAB tissue, revealing numerous fine-grained subregions, with clear spatial separation in UMAP space. These results collectively demonstrate robustness of our stUAI in identifying spatial domains across diverse tissues. Overall, we could also find that stUAI achieve state-of-the-art average performance, as shown in Figure 3. Additional results and analysis are provided in Appendix C and Appendix D.

### 3.3 ABLATION STUDY AND EVALUATION ON DIFFERENT FUSION STRATEGIES

To validate the effectiveness of different modules and fusion strategies in our proposed stUAI, we conducted experiments on the DLPFC (151671), HBC, and MAB tissue datasets.

**Ablation Study.** We evaluate several model variants to verify the contribution of each module as shown in Table 2. The results show that removing $\mathcal{L}_{\text{OTUA}}$ leads to a performance drop, underscoring the role of alignment in enhancing embedding semantics. Excluding $\mathcal{L}_{\text{CGCL}}$ further demonstrates the importance of contrastive learning, while removing $\mathcal{L}_{\text{ZINB\_reg}}$ highlights the critical role of spatial regularization, with ZINB modeling particularly suitable for the sparse and discrete nature of SRT data. Its absence results in a substantial decline in performance. Overall, these findings verify the effectiveness and necessity of all proposed modules. More ablation analyses are provided in Appendix E.

Table 2: Ablation study against several variants.

| Model | Metric | DLPFC | HBC | MAB |
|---|---|---|---|---|
| stUAI w/o $\mathcal{L}_{\text{CGCL}}$ | ARI | 0.62 | 0.57 | 0.46 |
| | NMI | 0.71 | 0.64 | 0.69 |
| stUAI w/o $\mathcal{L}_{\text{OTUA}}$ | ARI | 0.63 | 0.59 | 0.47 |
| | NMI | 0.71 | 0.65 | 0.70 |
| stUAI w/o $\mathcal{L}_{\text{ZINB\_reg}}$ | ARI | 0.68 | 0.61 | 0.47 |
| | NMI | 0.75 | 0.64 | 0.70 |
| stUAI (Ours) | ARI | **0.84** | **0.65** | **0.49** |
| | NMI | **0.78** | **0.69** | **0.71** |

**Fusion Strategies.** Here we test the performance of several different fusion strategies ($M_1$: $\mathbf{H}^{(f)}$ only, $M_2$: $\mathbf{H}^{(s)}$ only, $M_3$: $(\mathbf{H}^{(f)} + \mathbf{H}^{(s)})/2$, $M_4$: combine $\mathbf{H}^{(f)}$ and $\mathbf{H}^{(s)}$ via attention mechanism, and $M_5$: our uncertainty-aware fusion in Equation 5). As shown in Table 3, $M_1$ performs particularly poorly, which underscores the importance of spatial information for spatial clustering. $M_2$ performs worse than three fusion strategies $M_3$, $M_4$ and $M_5$, possibly because, in biological reality, cells of the same type are not always spatially adjacent. Comparing $M_3$ and $M_4$ reveals that the semantic knowledge weights contained in the two modalities are distinct, simply averaging them will cause one modality to be partially overwhelmed

Table 3: Comparison of different fusions.

| Model | Metric | DLPFC | HBC | MAB |
|---|---|---|---|---|
| $M_1$ | ARI | 0.23 | 0.48 | 0.43 |
| | NMI | 0.21 | 0.51 | 0.62 |
| $M_2$ | ARI | 0.71 | 0.60 | 0.46 |
| | NMI | 0.69 | 0.65 | 0.68 |
| $M_3$ | ARI | 0.75 | 0.61 | 0.45 |
| | NMI | 0.70 | 0.66 | 0.70 |
| $M_4$ | ARI | 0.78 | 0.62 | 0.47 |
| | NMI | 0.71 | 0.68 | 0.69 |
| $M_5$ (Ours) | ARI | **0.84** | **0.65** | **0.49** |
| | NMI | **0.78** | **0.69** | **0.71** |

by the other. Additionally, $M_3$ and $M_4$ effectively integrate feature modality and spatial modality information, but its performance is still inferior to our $M_5$, which is because our strategy captures uncertainties arising from technical or intrinsic data factors, mitigating their negative impact to ensure that representation components with higher uncertainty contribute less to the fused representation. More fusion analyses are provided in Appendix E.

## 3.4 SENSITIVITY ANALYSIS

In this section, we investigate the sensitivity of two key hyperparameters on DLPFC (151507). In Figure 4(a), $k$ is the number of neighbors of each spot in the feature graph. It can seen that when $k$ varies, our model stUAI performance first increases, peaks at $k = 14$, and then gradually decreases. The reason may be that when $k$ is small, increasing $k$ allows more relevant

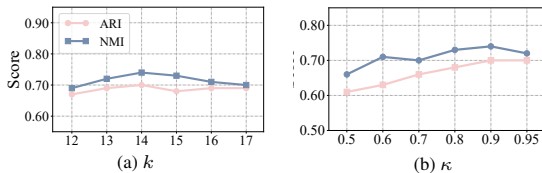

Figure 4: Sensitivity experimental results.

cells to be included, providing richer semantic information. However, when $k$ becomes too large, the inclusion of inherent noise may instead hinder the learning of accurate representations.

Next, in Figure 4(b), we analyze the sensitivity of the confidence coefficient $\kappa$ in positive sample selection for intra-view contrastive learning. It's known that a poor initial clustering may misguide subsequent iterations by reinforcing the flawed structure. To mitigate this issue, we introduce the confidence coefficient $\kappa$ to regulate the selection of positive samples. We vary $\kappa$ from 0.50 to 0.95, and the results show that the model achieved the best performance when $\kappa = 0.90$. It indicates that when $\kappa$ is relatively small, the model may include incorrect positive samples, whereas when $\kappa$ approaches 1, some true positive samples may be missed. Additional sensitivity analyses on different slices of DLPFC are provided in Appendix E.

## 3.5 UNCERTAINTY VISUALIZATION AND VERIFICATION ANALYSIS

SRT data often contain a large number of pseudo-zero values, meaning that the gene is indeed expressed but appears as zero because of factors such as errors in tissue processing or insufficient sequencing depth, which leads to data uncertainty. To demonstrate the rationale of quantifying uncertainty through distributional representation learning, we further conduct visualization experiments. In Figure 5, we visualize the number of dropouts (zero values) in gene expression and the uncertainty scores from the learned distributional representations of the spatial view for all spots on DLPFC (151507), where we extract the variance vector

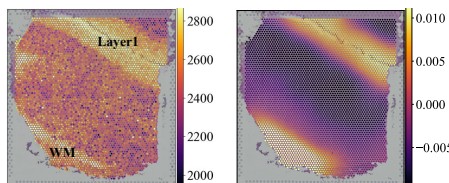

(a) True Dropouts    (b) Uncertainty Scores

Figure 5: Uncertainty analysis.

of each distribution and take its average as the uncertainty score for that spot. It can be observed that Figure 5(a) and (b) exhibit highly similar patterns, where both the highlighted regions are mainly concentrated in Layer 1 and WM. It indicates that the uncertainty quantified by our stUAI largely reflects the inherent uncertainty of the data, and thus carries certain biological significance.

## 3.6 VALIDATION ON DOWNSTREAM TASKS

**Detection of Differentially Expressed Genes (DEGs).** Based on the spatial clustering results, we compare the expression levels of each gene across different clusters and perform significance testing using t-test. Figures 6(a)-(b) show volcano plots for Layers 2 and 3 of the DLPFC (151507) and the detected DEGs, which is also consistent with the gene expression visualization in Figure 11 of Appendix F. Figures 6(c)-(d) also compare the log2FC of the top three DEGs between our stUAI and competitive baselines DUSTED and GraphST, showing that stUAI achieves higher log2FC values. It further validates that the spatial clustering from our stUAI exhibit more significant inter-cluster differences. Further detection and visualization of DEGs can be found in Appendix F.

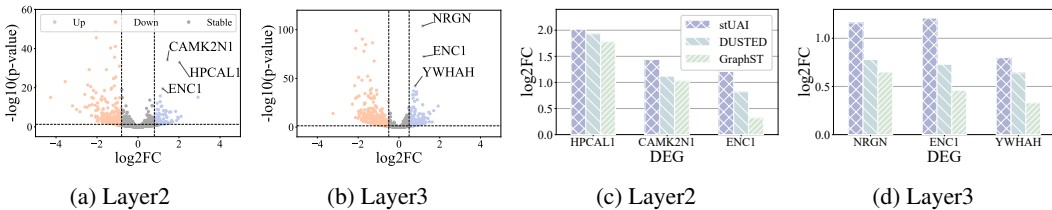

| (a) Layer2 | (b) Layer3 | (c) Layer2 | (d) Layer3 |

Figure 6: Quantitative analysis of differentially expressed genes (DEGs).

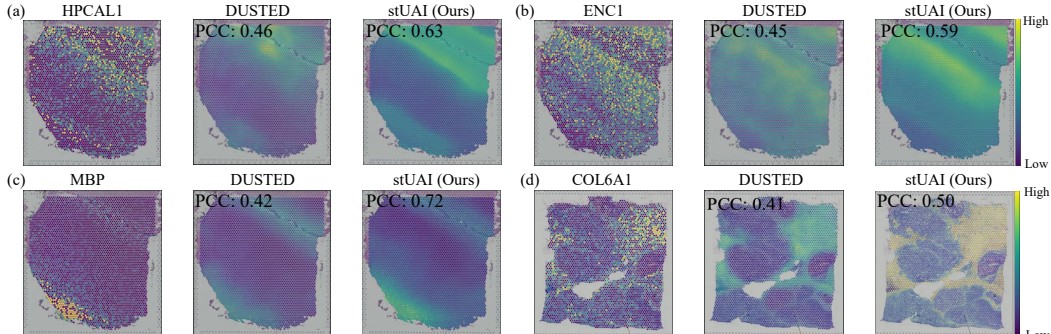

Figure 8: Comparison of gene imputation of DEGs on DLPFC and HBC datasets.

**Visualization of Pseudotime Trajectory Inference.** We leverage the learned representations and apply the DPT algorithm (Haghverdi et al., 2016) to conduct pseudotime trajectory inference, thereby exploring the developmental pathways of the tissues. In Figure 7, we visualize the pseudotime trajectories obtained on the DLPFC (151671) dataset using SCANPY, DUSTED, and our stUAI. As shown, stUAI produces clearer hierarchical trajectory structures and smoother color transitions. The trajectory inferred from stUAI also shows a stronger alignment with the manual annotation compared with SCANPY and DUSTED, and it originates from WM and progresses toward Layer 3, which is consistent with widely recognized biological facts and observations.

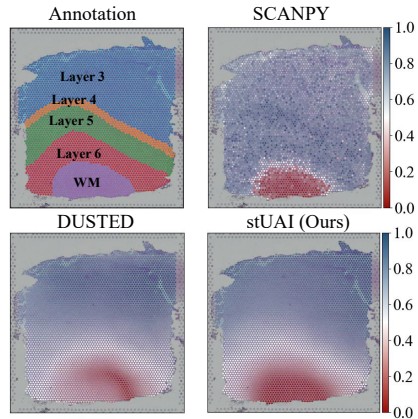

Figure 7: Pseudotime trajectory.

**Application on Gene Imputation.** To mitigate the sparsity and high noise of SRT data, we perform gene imputation to recover the gene expressions. Specifically, for the DEGs HPCAL1, ENC1, MBP of DLPFC (151507) and COL6A1 of the HBC dataset, we randomly mask 30% of their expressions and then perform gene imputation using stUAI and DUSTED. Imputation accuracy is assessed using the Pearson correlation coefficient (PCC, Pearson, 1894) with reference to the ground truth. As shown in Figure 8, stUAI achieves more accurate gene imputation with higher PCC values, which also demonstrates the success of our stUAI. Additional analysis about denoising is provided in Appendix G.

**Gene Pathway Enrichment Analysis.** We further conduct KEGG enrichment analysis on the cluster located at the bottom left of the HBC dataset identified by stUAI in Figure 2(a) (mostly corresponding to IDC_4 in Figure 17(b)), where we first identify the DEGs associated with this cluster and then query the pathways (e.g., metabolic, signaling, and disease pathways) by hypergeometric

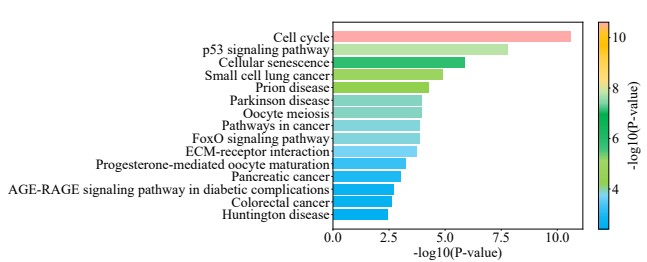

Figure 9: KEGG Enrichment Result.

enrichment test based on these DEGs. As shown in Figure 9, the cell cycle and p53 signaling pathways exhibit the lowest p-values, indicating their strong enrichment and functional relevance within the domain under consideration. The coordinated dysregulation of the cell cycle and p53 pathways is well known to drive malignant progression. The inactivation of p53, the guardian of the genome, leads to genomic instability and removes the brakes on cell-cycle control, while the sustained activation of the cell cycle pathway subsequently enables uncontrolled cellular proliferation. Together, these alterations form the fundamental basis of IDC's invasive growth and poor clinical prognosis, which indicates that the enrichment result of our stUAI align with the biological knowledge (Waldemer-Streyer et al., 2017), thereby strongly supporting the biological relevance of the clusters identified by our stUAI.

## 4 CONCLUSION

In this work, we propose a novel and powerful Uncertainty-Aware Integration framework stUAI to tackle key challenges in SRT, including expression sparsity, over-dispersion, and modality-specific heterogeneity. By constructing spatial and gene-expression graphs, stUAI learns view-specific embeddings through a dual-channel GCN and models them as Gaussian distributions to quantify uncertainty. This uncertainty guides both cluster-guided contrastive learning and cross-view alignment via optimal transport. Additionally, a zero-inflated negative binomial decoder mitigates dropout effects, and spatial structural constraints ensure continuity of spatial patterns. Extensive experiments on multiple benchmark datasets demonstrate that stUAI consistently improves spatial clustering, offering a powerful solution for spatial omics analysis. Furthermore, comprehensive biological analyses and interpretations provide deeper insights into the underlying mechanisms.

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

# A  Pseudo-Code of Our Framework

The pseudo-code of our proposed stUAI is shown in Algorithm 1. The source code is available for reproducibility at: https://anonymous.4open.science/r/stUAI-FF15/.

---

**Algorithm 1** The Optimization Algorithm of stUAI

---

**Input**: Spatial positions $\mathbf{S} = \{(x_i, y_i)\}_{i=1}^{N}$; Gene expression matrix $\mathbf{F} \in \mathbb{R}^{N \times D}$; Maximum number of iterations $I_{\max}$;

**Output**: Clustering result $\mathcal{C}_1, \ldots, \mathcal{C}_K$;

$\backslash \mathtt{j} \in [1 : \mathtt{N}], \mathtt{i} \in \{\mathtt{s}, \mathtt{f}\} \backslash$

Construct the spatial graph $\mathcal{G}^{(\mathrm{s})} = (\mathbf{A}^{(\mathrm{s})}, \mathbf{F})$ and feature graph $\mathcal{G}^{(\mathrm{f})} = (\mathbf{A}^{(\mathrm{f})}, \mathbf{F})$;

Initialize the trainable parameters in GCN encoders, the mean and variance networks $f_{\mathrm{m}}^{(i)}$ and $f_{\mathrm{v}}^{(i)}$, the ZINB-based decoder networks $f_{\mathrm{D}}^1, f_{\mathrm{D}}^2, f_{\mathrm{D}}^3$, and the index sets $\mathcal{I}_1, \ldots, \mathcal{I}_N$ of positive nodes;

Set $t = 0$;

**while** $t \leq I_{\max}$ **do**

  Update latent embeddings $\mathbf{H}^{(\mathrm{s})}$ and $\mathbf{H}^{(\mathrm{f})}$ by GCN encoders;

  $\backslash \mathtt{Compute}\ \mathtt{L_{CGCL}} \backslash$

  Update the distributional representations in Equation 1;

  Sample $\mathbf{h}_j^{(i),1}, \mathbf{h}_j^{(i),2}$ by Equation 2 from the distributional representations;

  Perform representation fusion by Equation 5 and cluster based on fused spot representations;

  Leverage intermediate clustering results to update $\mathcal{I}_1, \ldots, \mathcal{I}_N$ and construct constractive pairs;

  Compute the clustering-guided contrastive learning loss $\mathcal{L}_{\mathrm{CGCL}}$ in Equation 3;

  $\backslash \mathtt{Compute}\ \mathcal{L}_{\mathtt{OTUA}} \backslash$

  Compute the OT-based uncertainty alignment loss $\mathcal{L}_{\mathrm{OTUA}}$ in Equation 4;

  $\backslash \mathtt{Compute}\ \mathcal{L}_{\mathtt{ZINB\_reg}} \backslash$

  Update the parameter estimators of the ZINB distribution in Equation 6;

  Compute the ZINB-based reconstruction loss with spatial regularization $\mathcal{L}_{\mathrm{ZINB\_reg}}$ in Equation 7;

  $\backslash \mathtt{Compute}\ \mathcal{L}_{\mathtt{Total}} \backslash$

  Compute the total loss $\mathcal{L}_{\mathrm{Total}}$ in Equation 8;

  $\backslash \mathtt{Optimization} \backslash$

  Conduct back-propagation and update the whole network in stUAI by minimizing $\mathcal{L}_{\mathrm{Total}}$;

  Set $t = t + 1$;

**end while**

$\backslash \mathtt{Final\ Clustering} \backslash$

**Return**: Generate k-means clustering result $\mathcal{C}_1, \ldots, \mathcal{C}_K$ based on the final integrated spot representations.

---

# B  Related Work

## B.1  GNN for Spatial Transcriptomics Clustering

Graph deep learning has gained significant attention in SRT clustering due to its ability to model complex relationships between spatial locations and gene expression data. Along this line, a variety of GNN-based methods have been developed to integrate spatial and gene expression information for SRT clustering. These approaches typically aim to capture spatial organization and transcriptional similarity in tissue sections, while addressing challenges such as multi-modal data fusion, spatial continuity, and tissue heterogeneity. For example, SpaGCN (Hu et al., 2021) incorporates histological context alongside spatial and transcriptomic data using a unified GCN framework to identify spatial domains and detect domain-specific gene expression patterns. SEDR (Xu et al., 2024) integrates gene expression and spatial coordinates via a combination of auto-encoders and variational GNNs. Similarly, CCST (Li et al., 2022) leverages GCNs to adaptively model spatial relationships, enabling accurate unsupervised clustering and discovery of cell subtypes and states in SRT data. In addition, STAGATE(Dong & Zhang, 2022b) is a graph attention auto-encoder that integrates spatial and gene expression data to accurately identify spatial domains and improve clus-

tering while preserving expression patterns. Extending this line of work, DeepST (Xu et al., 2022) integrates image, gene expression, and spatial data using GNN and denoising autoencoders, while employing domain adversarial networks for batch integration for SRT. More recently, MAFN (Zhu et al., 2024) leverages dual GCN encoders and cross-view attention to adaptively fuse spatial and gene expression features for accurate clustering of SRT data. DUSTED (Zhu et al., 2025) presents a dual-attention graph autoencoder for spatial transcriptomics, denoising gene expression by modeling spatial features and noise variability. Spotscape (Oh et al., 2025) addresses the issue of poor representations for boundary spots by capturing global context and scaling similarities to integrate multiple slices. Other methods, such as Spatial-MGCN and SCGDL, adopt multi-view or hierarchical learning strategies to better fuse complementary spatial and molecular modalities (Ren et al., 2022; Long et al., 2023). While these methods have shown strong performance, they typically focus on data integration and representation learning without explicitly accounting for the uncertainty introduced by noise and modality-specific variability. In contrast, our proposed stUAI emphasizes the importance of uncertainty modeling, enabling more robust clustering through uncertainty-aware fusion of spatial and gene expression information.

### B.2 UNCERTAINTY-BASED DEEP LEARNING

Uncertainty in deep learning (Blundell et al., 2015; Gawlikowski et al., 2023) has become a critical factor in building trustworthy AI systems. Recent studies have explored various modeling paradigms to quantify and mitigate uncertainty, which are typically grouped based on how they handle different types of uncertainty. *Aleatoric uncertainty* reflects the intrinsic variability in the data—caused by factors like experimental noise, dynamic environments, or biological heterogeneity. Such uncertainty is typically captured by modeling input-related noise, often using a Gaussian distribution (Vilnis & McCallum, 2015; Yang et al., 2021; Gao et al., 2024). *Epistemic uncertainty*, by contrast, stems from limited knowledge of the underlying data distribution or insufficient model expressiveness. Bayesian deep learning, especially Bayesian neural networks (BNNs), address this by imposing priors on model parameters and conducting posterior inference to represent uncertainty over the model itself (Goan & Fookes, 2020; Sun et al., 2019; Pearce et al., 2020). *Hybrid approaches* seek to simultaneously capture both aleatoric and epistemic uncertainty, providing a more holistic uncertainty estimate (Hofer et al., 2002; Jiang et al., 2018; Li et al., 2020). An alternative line of research involves *evidence-based methods*, which utilize the Dempster-Shafer theory (Dempster, 1967; 1968) to model uncertainty without requiring strong probabilistic assumptions. These methods can represent uncertainty and conflict explicitly (Sensoy et al., 2018; Han et al., 2020), and have been applied in tasks like multi-modal sensor fusion and OOD detection. Lastly, *ensemble-based techniques* estimate uncertainty by combining predictions from multiple independently trained models (Lakshminarayanan et al., 2017; Rahaman et al., 2021). Unlike prior works that largely overlook modality-specific uncertainty, our framework focuses on modeling uncertainty to guide the integration of spatial position and gene expression information for robust spatial clustering.

## C PERFORMANCE OF SPATIAL CLUSTERING FOR OTHER DATASETS

Table 4: Performance of spatial clustering on HBC dataset and MAB dataset. The best result in all the methods are highlighted with **bold**.

| Dataset | Metric | SCANPY | GraphST | SCGDL | stLearn | DUSTED | stUAI (Ours) |
|---------|--------|--------|---------|-------|---------|--------|--------------|
| **HBC** | ARI | 0.49 | 0.53 | 0.35 | 0.55 | 0.57 | **0.65** |
| | NMI | 0.52 | 0.67 | 0.43 | 0.63 | 0.63 | **0.69** |
| **MAB** | ARI | 0.23 | 0.41 | 0.26 | 0.38 | 0.40 | **0.49** |
| | NMI | 0.45 | **0.71** | 0.64 | 0.66 | 0.65 | **0.71** |

To further evaluate the performance of our stUAI, we conduct experiments on two additional datasets, as shown in Table 4. From the results, we observe the following: First, overall, all algorithms perform worse on the HBC and MAB datasets compared to the DLPFC dataset, which may be due to the more complex spatial structure of the HBC and MAB datasets, with more clusters that are harder to represent. Second, the conclusion that spatially-aware methods perform better still holds, indicating that spatial information plays a facilitating role in capturing heterogeneity. Finally,

our stUAI still maintains excellent performance on the HBC (ARI = 0.65, NMI = 0.69) and MAB (ARI = 0.49, NMI = 0.71) datasets, highlighting the superiority of our approach.

# D  PERFORMANCE OF SPATIAL CLUSTERING UNDER UNSUPERVISED METRICS

Table 5: Performance comparison under Jaccard similarity coefficient and Moran's I score across three datasets. The best result in all the methods are highlighted with **bold**.

| Dataset | Metric | SCANPY | STAGATE | GraphST | DUSTED | stUAI (Ours) |
|---|---|---|---|---|---|---|
| **DLPFC (151507)** | Jaccard | 0.11 | 0.49 | 0.37 | 0.40 | **0.62** |
| | Moran's I | 0.47 | 0.86 | 0.77 | 0.66 | **0.91** |
| **HBC** | Jaccard | 0.19 | 0.31 | 0.40 | 0.43 | **0.47** |
| | Moran's I | 0.56 | 0.77 | 0.76 | 0.62 | **0.85** |
| **MAB** | Jaccard | 0.14 | 0.25 | 0.23 | 0.22 | **0.33** |
| | Moran's I | 0.50 | 0.74 | 0.59 | 0.64 | **0.75** |

Since ground-truth annotations are often unavailable in many scenarios, it is essential to evaluate the quality of our method for spatial clustering using various and unsupervised metrics. Hence we evaluate our stUAI and competing baselines on three benchmark datasets using two additional metrics: Jaccard similarity, which measures the reproducibility of domain assignments under subsampling, and Moran's I, which unsupervisedly quantifies spatial autocorrelation. As shown in Table 5, we can find the following observations: (i) Spatially-aware methods consistently outperform non-spatial ones, with approaches like SCANPY showing severe degradation in spatial coherence; (ii) Our stUAI achieves the best overall results, including Jaccard improvements on three datasets and a markedly higher Moran's I on DLPFC and HBC, demonstrating its robustness and effectiveness in capturing biologically meaningful spatial domains.

# E  ADDITIONAL ANALYSIS ON ABLATION STUDY AND SENSITIVITY ANALYSIS

Table 6: Ablation study against additional variants. The best results are highlighted with **bold**.

| Dataset | Metric | $V_1$ | $V_2$ | $V_3$ | $V_4$ | $V_5$ | $V_6$ | $V_7$ | stUAI (Ours) |
|---|---|---|---|---|---|---|---|---|---|
| **DLPFC** | ARI | 0.61 | 0.30 | 0.52 | 0.66 | 0.60 | 0.81 | 0.80 | **0.84** |
| **(151671)** | NMI | 0.69 | 0.22 | 0.66 | 0.72 | 0.68 | 0.74 | 0.75 | **0.78** |
| **HBC** | ARI | 0.58 | 0.45 | 0.58 | 0.57 | 0.56 | 0.62 | 0.60 | **0.65** |
| | NMI | 0.63 | 0.52 | 0.62 | 0.63 | 0.62 | 0.65 | 0.64 | **0.69** |
| **MAB** | ARI | 0.45 | 0.44 | 0.46 | 0.43 | 0.45 | 0.46 | 0.44 | **0.49** |
| | NMI | 0.68 | 0.63 | 0.67 | 0.66 | 0.67 | 0.66 | 0.67 | **0.71** |

Here, we explore more variants to validate the effectiveness and necessity of each module in our method: (i) $V_1$: solely adopt the spatial-level graph (*i.e.*, $\mathcal{L} = \mathcal{L}_{\text{CGCL}}^{(s)} + \mathcal{L}_{\text{ZINB\_reg}}$); (ii) $V_2$: solely adopt the feature-level graph (*i.e.*, $\mathcal{L} = \mathcal{L}_{\text{CGCL}}^{(f)} + \mathcal{L}_{\text{ZINB\_reg}}$); (iii) $V_3$: solely adopt cross-view distributional representation alignment (*i.e.*, $\mathcal{L} = \mathcal{L}_{\text{OTUA}}$); (iv) $V_4$: solely adopt cluster-guided intra-view contrastive learning (*i.e.*, $\mathcal{L} = \mathcal{L}_{\text{CGCL}}$); (v) $V_5$: solely adopt ZINB-based reconstruction (*i.e.*, $\mathcal{L} = \mathcal{L}_{\text{ZINB\_reg}}$); (vi) $V_6$: replace cluster-guided contrastive learning with standard contrastive learning; (vii) $V_7$: replace ZINB-based reconstruction with MSE reconstruction.

The comparative results are summarized in Table 6. Compared to $V_1$ and $V_2$, we can observe that spatial information plays a more critical role than feature information in domain identification tasks. When comparing $V_3$, $V_4$, $V_5$ with our stUAI, it becomes evident that preserving any single module alone cannot achieve ideal performance, reaffirming that all proposed modules are indispensable. The comparison between $V_6$ and our method demonstrates that the positive samples selected based on clustering confidence provide superior and more stable supervisory signals for contrastive learning. Furthermore, comparing $V_7$ with our approach reveals that MSE fails to adequately capture the

sparse and over-dispersion nature of single-cell sequencing data compared to ZINB, consequently leading to performance degradation.

**Ablation studies on different slices of DLPFC.** Furthermore, to comprehensively evaluate the effectiveness and model robustness of our method on the DLPFC dataset, we conduct ablation experiments and fusion strategy tests on different slices of DLPFC. First, we perform ablation studies corresponding to Table 2 in Section 3.3 and Table 6 on different slices of DLPFC. The results are shown in Table 7. It can be observed that, under the ARI and NMI metrics, our stUAI achieves optimal performance on most slices while also exhibiting the best average performance across all slices. It demonstrates that each component of stUAI plays a crucial role in improving the performance of domain identification on each slice of DLPFC.

Table 7: Ablation study against different variants on all slices of the DLPFC dataset. The best results are highlighted with **bold**.

(a)

| Model | Metric | 151507 | 151508 | 151509 | 151510 | 151669 | 151670 | 151671 | 151672 | 151673 | 151674 | 151675 | 151676 | Average |
|---|---|---|---|---|---|---|---|---|---|---|---|---|---|---|
| stUAI w/o $\mathcal{L}_{\text{CGCL}}$ | ARI | 0.67 | **0.67** | 0.52 | 0.68 | 0.48 | 0.43 | 0.62 | 0.75 | 0.53 | 0.53 | 0.56 | 0.53 | 0.58 |
| | NMI | 0.71 | **0.72** | 0.65 | 0.68 | 0.54 | 0.54 | 0.71 | 0.71 | 0.65 | 0.63 | 0.64 | 0.63 | 0.65 |
| stUAI w/o $\mathcal{L}_{\text{OTUA}}$ | ARI | 0.57 | 0.54 | 0.50 | 0.51 | 0.38 | **0.50** | 0.63 | 0.73 | **0.57** | 0.50 | 0.51 | 0.51 | 0.54 |
| | NMI | 0.71 | 0.66 | 0.64 | 0.66 | 0.50 | 0.59 | 0.71 | 0.70 | **0.66** | 0.62 | 0.65 | 0.66 | 0.65 |
| stUAI w/o $\mathcal{L}_{\text{ZINB\_reg}}$ | ARI | 0.60 | 0.61 | 0.45 | 0.50 | 0.46 | 0.40 | 0.68 | 0.73 | 0.51 | 0.53 | 0.51 | 0.41 | 0.53 |
| | NMI | 0.68 | 0.65 | 0.63 | 0.61 | 0.56 | 0.51 | 0.75 | 0.72 | 0.65 | 0.58 | 0.62 | 0.59 | 0.63 |
| stUAI (Ours) | ARI | **0.70** | 0.65 | **0.58** | **0.70** | **0.50** | 0.49 | **0.84** | **0.79** | 0.56 | **0.55** | **0.58** | **0.55** | **0.62** |
| | NMI | **0.74** | 0.70 | **0.66** | **0.72** | **0.60** | **0.61** | **0.78** | **0.76** | **0.66** | **0.65** | **0.67** | **0.68** | **0.69** |

(b)

| Model | Metric | 151507 | 151508 | 151509 | 151510 | 151669 | 151670 | 151671 | 151672 | 151673 | 151674 | 151675 | 151676 | Average |
|---|---|---|---|---|---|---|---|---|---|---|---|---|---|---|
| $\mathbf{V}_1$ | ARI | 0.54 | 0.51 | 0.51 | 0.53 | 0.40 | 0.37 | 0.61 | 0.61 | 0.44 | 0.48 | 0.45 | 0.42 | 0.49 |
| | NMI | 0.68 | 0.66 | 0.62 | 0.65 | 0.52 | 0.47 | 0.69 | 0.63 | 0.48 | 0.51 | 0.53 | 0.58 | 0.59 |
| $\mathbf{V}_2$ | ARI | 0.22 | 0.16 | 0.25 | 0.23 | 0.13 | 0.13 | 0.30 | 0.35 | 0.30 | 0.31 | 0.28 | 0.23 | 0.24 |
| | NMI | 0.27 | 0.23 | 0.31 | 0.33 | 0.17 | 0.19 | 0.22 | 0.28 | 0.42 | 0.39 | 0.39 | 0.33 | 0.29 |
| $\mathbf{V}_3$ | ARI | 0.60 | 0.61 | 0.52 | 0.50 | 0.46 | 0.40 | 0.52 | 0.66 | 0.51 | 0.44 | 0.51 | 0.41 | 0.51 |
| | NMI | 0.68 | 0.65 | 0.62 | 0.61 | 0.50 | 0.41 | 0.66 | 0.72 | 0.55 | 0.48 | 0.59 | 0.59 | 0.59 |
| $\mathbf{V}_4$ | ARI | 0.53 | 0.54 | 0.54 | 0.60 | 0.45 | 0.43 | 0.66 | 0.66 | 0.48 | 0.47 | 0.47 | 0.42 | 0.52 |
| | NMI | 0.69 | 0.67 | 0.57 | 0.66 | 0.46 | 0.54 | 0.72 | 0.69 | 0.56 | 0.56 | 0.59 | 0.61 | 0.61 |
| $\mathbf{V}_5$ | ARI | 0.56 | 0.54 | 0.50 | 0.51 | 0.37 | 0.40 | 0.60 | 0.63 | 0.52 | 0.46 | 0.47 | 0.40 | 0.50 |
| | NMI | 0.70 | 0.66 | 0.55 | 0.66 | 0.57 | 0.49 | 0.68 | 0.65 | 0.58 | 0.54 | 0.55 | 0.56 | 0.60 |
| $\mathbf{V}_6$ | ARI | 0.62 | **0.67** | 0.56 | 0.61 | 0.47 | 0.44 | 0.81 | 0.76 | 0.51 | 0.48 | 0.51 | 0.49 | 0.58 |
| | NMI | 0.70 | 0.65 | 0.60 | 0.61 | 0.52 | 0.57 | 0.74 | 0.69 | 0.57 | 0.61 | 0.64 | **0.69** | 0.63 |
| $\mathbf{V}_7$ | ARI | 0.64 | 0.63 | 0.54 | 0.60 | 0.45 | **0.51** | 0.80 | 0.74 | 0.46 | 0.42 | 0.47 | 0.53 | 0.57 |
| | NMI | 0.73 | **0.72** | 0.59 | **0.73** | 0.53 | **0.61** | 0.75 | 0.71 | 0.49 | 0.57 | 0.58 | 0.64 | 0.64 |
| stUAI (Ours) | ARI | **0.70** | 0.65 | **0.58** | **0.70** | **0.50** | 0.49 | **0.84** | **0.79** | **0.56** | **0.55** | **0.58** | **0.55** | **0.62** |
| | NMI | **0.74** | 0.70 | **0.66** | 0.72 | **0.60** | 0.61 | **0.78** | **0.76** | **0.66** | **0.65** | **0.67** | 0.68 | **0.69** |

Table 8: Comparison of different fusions on all slices of the DLPFC dataset. The best results are highlighted with **bold**.

| Model | Metric | 151507 | 151508 | 151509 | 151510 | 151669 | 151670 | 151671 | 151672 | 151673 | 151674 | 151675 | 151676 | Average |
|---|---|---|---|---|---|---|---|---|---|---|---|---|---|---|
| $\mathbf{M}_1$ | ARI | 0.21 | 0.14 | 0.22 | 0.21 | 0.11 | 0.12 | 0.23 | 0.31 | 0.26 | 0.29 | 0.26 | 0.21 | 0.21 |
| | NMI | 0.26 | 0.22 | 0.30 | 0.35 | 0.14 | 0.15 | 0.21 | 0.27 | 0.37 | 0.36 | 0.36 | 0.30 | 0.27 |
| $\mathbf{M}_2$ | ARI | 0.58 | 0.56 | 0.60 | 0.64 | 0.45 | 0.46 | 0.71 | 0.76 | 0.48 | 0.50 | 0.54 | 0.51 | 0.57 |
| | NMI | 0.70 | 0.68 | 0.60 | 0.69 | 0.52 | 0.52 | 0.69 | 0.73 | 0.56 | 0.62 | 0.63 | 0.64 | 0.63 |
| $\mathbf{M}_3$ | ARI | 0.58 | 0.58 | **0.61** | 0.60 | 0.46 | 0.39 | 0.75 | 0.68 | 0.52 | 0.51 | 0.56 | **0.56** | 0.57 |
| | NMI | 0.71 | 0.62 | 0.64 | 0.70 | 0.55 | 0.55 | 0.70 | 0.66 | 0.61 | 0.62 | **0.68** | 0.62 | 0.64 |
| $\mathbf{M}_4$ | ARI | 0.62 | 0.63 | 0.60 | 0.58 | **0.51** | 0.44 | 0.78 | 0.73 | **0.57** | 0.50 | 0.41 | 0.53 | 0.58 |
| | NMI | 0.69 | **0.72** | 0.63 | 0.63 | 0.57 | 0.59 | 0.71 | 0.71 | 0.64 | 0.63 | 0.60 | 0.64 | 0.65 |
| stUAI (Ours) | ARI | **0.70** | **0.65** | 0.58 | **0.70** | 0.50 | **0.49** | **0.84** | **0.79** | 0.56 | **0.55** | **0.58** | 0.55 | **0.62** |
| | NMI | **0.74** | 0.70 | **0.66** | **0.72** | **0.60** | **0.61** | **0.78** | **0.76** | **0.66** | **0.65** | 0.67 | **0.68** | **0.69** |

**Fusion strategies on different slices of DLPFC.** Moreover, the performance comparison of different fusion strategies across various slices is shown in Table 8. It can be seen that on different slices of DLPFC, the spatial view consistently plays a more significant role in identification perfor-

mance. Additionally, the weighted fusion strategy yields better results than simple averaging, and our inverse-uncertainty weighting achieves superior performance compared to learnable attention weighting. Our method also demonstrates the best average identification performance across different slices. It further demonstrates the effectiveness and robustness of our designed fusion strategy.

**Sensitivity analysis on different slices of DLPFC.** Finally, we conduct sensitivity analyses on the hyperparameters $k$ (the number of neighbors for each spot in the feature graph) and $\kappa$ (confidence coefficient in positive sample selection) across various slices of DLPFC dataset. The results are shown in Figure 10. Evidently, the conclusions drawn from the hyperparameter analysis on DLPFC (151507) in Section 3.4 remain applicable to most

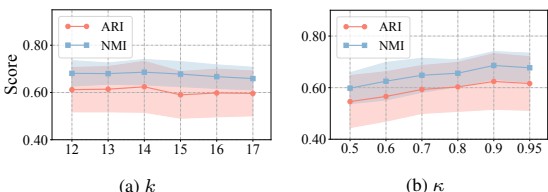

Figure 10: Sensitivity analysis on all slices of the DLPFC dataset.

other slices. Therefore, our hyperparameter settings, i.e., $k = 14, \kappa = 0.9$, are proven to be highly suitable across different slices.

## F  ADDITIONAL DEGs VISUALIZATION AND VERIFICATION ANALYSIS

To further validate the clustering performance of our stUAI, we conduct extensive DEGs analysis on DLPFC dataset and HBC dataset. The results show that spatial domains identified by our stUAI are consistent with biological knowledge, highlighting the superiority of stUAI.

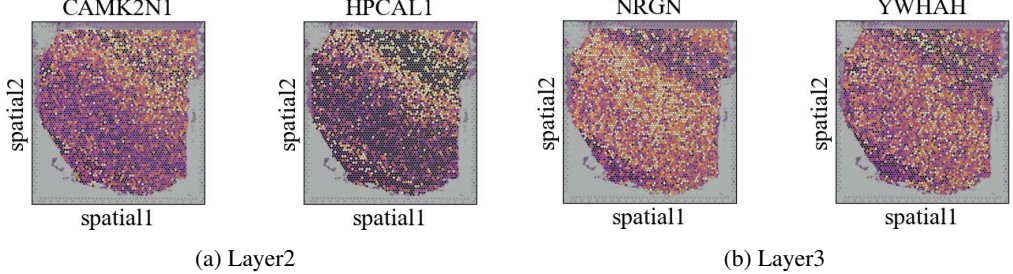

Figure 11: Experession visualization of DEGs in (a) Layer 2 and (b) Layer3 of DLPFC (151507).

**DEGs Analysis on DLPFC dataset.** We further conduct a comprehensive DEGs analysis on the DLPFC dataset to explore gene expression patterns across the six-layered structure of the human brain's gray matter. Drawing on established biological knowledge, the human brain's gray matter is known to have a complex six-layered structure, each of which plays a distinct role in cognitive and sensory processing. To gain deeper insights into this structure, focusing on DLPFC (151507) dataset, we begin by visualizing the DEGs across all six layers of the human brain gray matter using a volcano plot, which not only reveals the overall expression patterns in each layer but also enables us to identify the top three genes with the highest expression levels in each of the six layers, as illustrated in Figure 12. These genes are crucial for understanding the functional differences between the layers and their roles in neural processes. To further analyze these patterns, we provide a more detailed visualization of the gene expression of DEGs specifically in Layer 2 and Layer 3. Figure 11 presents a detailed visualization of the gene expression patterns in these two layers, highlighting the unique gene signatures that define their cellular and functional characteristics.

**DEGs Analysis on HBC dataset.** We further conduct DEGs analysis on HBC dataset. The visualization of DEGs using the violin plot is shown in Figure 13. The comparison between the healthy region Healthy_1 and the breast cancer malignant region IDC_4 reveals that the relative expression of the gene CXCL14 is significantly higher in the IDC_4 region, while its expression in Healthy_1 is not prominent, as shown in Figure 14. According to recent advancements in cancer research, CXCL14 has been identified as a crucial prognostic marker for breast cancer, as highlighted in the study by Waldemer-Streyer et al. (2017). This finding has been corroborated by various biological studies,

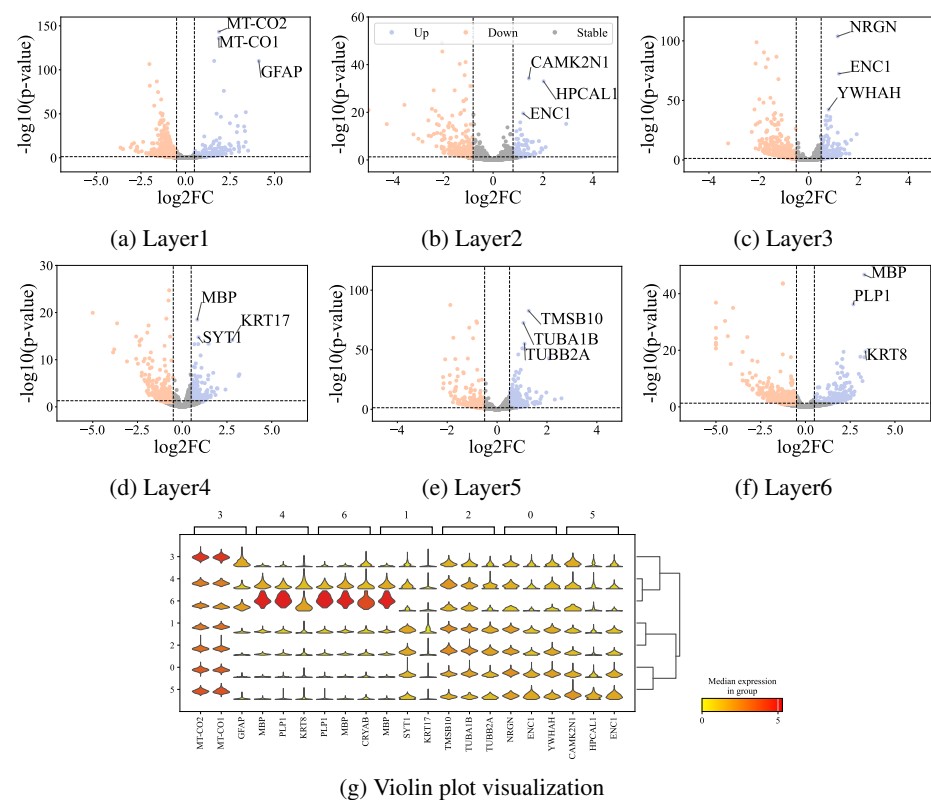

(a) Layer1     (b) Layer2     (c) Layer3

(d) Layer4     (e) Layer5     (f) Layer6

(g) Violin plot visualization

Figure 12: DEGs visualization analysis of six cerebral gray matter layers on DLPFC (151507).

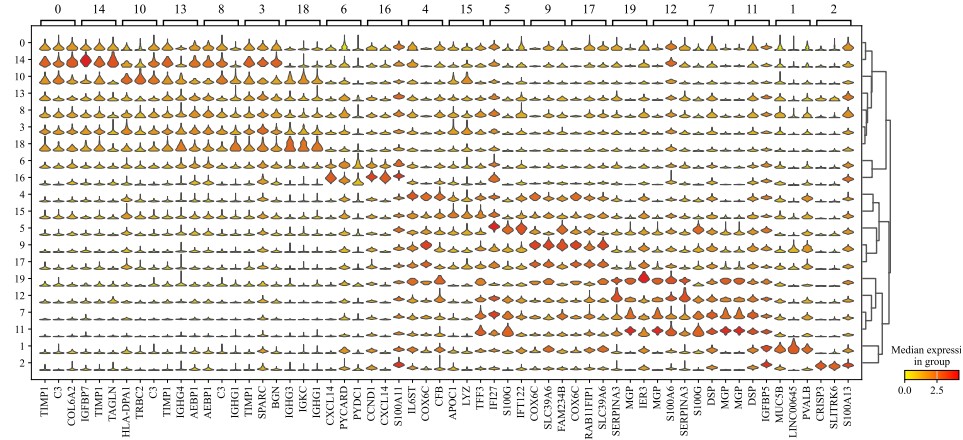

Figure 13: Violin plot visualization of DEGs on HBC dataset.

which suggest that CXCL14 plays a significant role in the tumor microenvironment, influencing cancer cell behavior and response to therapy. To further explore this, we visualize the volcano plot for DEG detection on HBC and calculate its PCC value of gene imputation, as depicted in Figure 15. The results of our stUAI show a strong correlation with the existing biological literature, reaffirming the relevance of CXCL14 as a potential biomarker for breast cancer prognosis. This consistency between our findings and prior research underscores the biological interpretability of our approach.

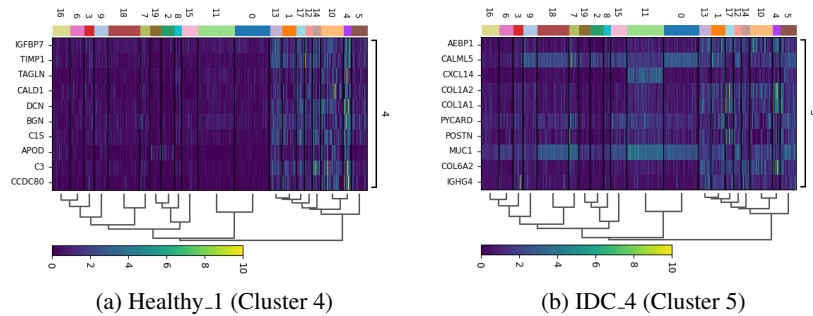

| (a) Healthy_1 (Cluster 4) | (b) IDC_4 (Cluster 5) |

Figure 14: Heatmap of the expressions on the top 10 differentially expressed genes (DEGs) between Healthy_1 and IDC_4 on HBC dataset.

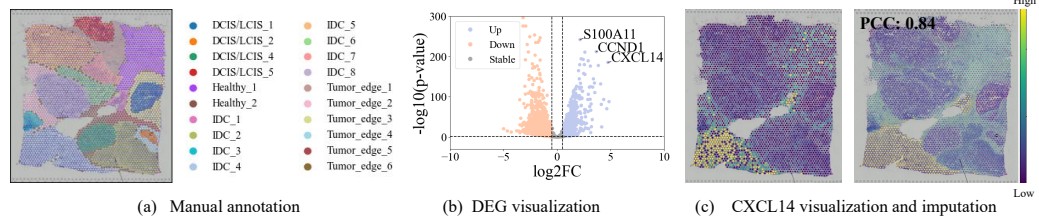

(a) Manual annotation     (b) DEG visualization     (c) CXCL14 visualization and imputation

Figure 15: (a) Visualization of manual annotation on HBC dataset; (b) Volcano plot for DEG detection; (c) Visualization and gene imputation for CXCL14.

## G  PERFORMANCE EVALUATION ON DENOISING

SRT data are inherently noisy and prone to dropout events, which can obscure biologically meaningful spatial patterns. We select six well-established layer marker genes (ATP2B4, FKBP1A, CRYM, NEFH, RXFP1, B3GALT2) in slice 151507 of the DLPFC and compare their raw spatial expression with the denoised expression generated by stUAI, as illustrated in Figure 16. The results show that the spatial expression patterns after denoising are more consistent with the spatial domains. These results suggest that our proposed stUAI not only captures coherent spatial domains but also enhances biologically relevant spatial expression signals.

## H  DISCUSSION ON THE LEARNED UNCERTAINTY

In our stUAI framework, we utilize the learned uncertainty for representation fusion, where we adopt an inverse-variance weighting strategy to downweight samples with high uncertainty, thereby reducing the negative influence of uncertainty and achieving reliable, stable, and more robust and interpretable representations. Consequently, when two views are fused, spots associated with higher uncertainty in one view receive smaller weights, mitigating the impact of highly uncertain components on the final domain identification result. In addition, one could consider removing spots that exhibit abnormally high uncertainty in both views before performing clustering, thereby reducing the risk of forming spurious clusters. Uncertainty can also be used to weight each spot's expression values during differential expression analysis, improving the reliability of detected differential expression genes. Moreover, uncertainty visualization, similar to Figure 5, can assist pathologists or biologists in assessing which regions yield trustworthy analytical results and can be overlaid with H&E images for visual validation. These directions provide promising opportunities for further investigation, and we will explore them in future work.

## I  DETAILED SETTING FOR EXPERIMENTS

**Evaluation Metrics.** To comprehensively evaluate spatial clustering performance, we adopt four widely used metrics. (i) *Supervised metrics*: Adjusted Rand Index (ARI) (Rand, 1971), Normalized

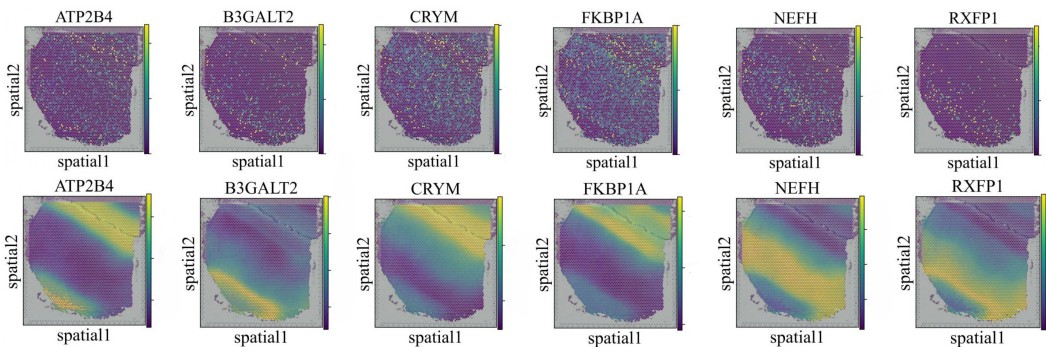

Figure 16: Spatial expressions of marker genes before and after denoising on DLPFC (151507).

Mutual Information (NMI) (Knops et al., 2006), and Jaccard Similarity Coefficient (Levandowsky & Winter, 1971). These metrics quantify the alignment between the predicted clustering assignments and the ground-truth annotations. (ii) *Unsupervised metrics*: Moran's I Score (Moran, 1950), which specifically quantifies spatial autocorrelation. Higher scores on these four metrics consistently reflect superior clustering quality.

**Datasets.** We conduct extensive experiments on three benchmark datasets, briefly introduced as follows. The first dataset is derived from the LIBD study of the human dorsolateral prefrontal cortex (DLPFC) (Maynard et al., 2021). It includes expert-curated annotations by Maynard et al. (2021), delineating six distinct cortical layers alongside the white matter (WM). Each slice in this dataset profiles 3460 to 4789 spots, 33,538 genes and spans between 5 and 7 spatial regions. The second dataset comes from the 10x Visium platform, focusing on human breast cancer (HBC) samples (Buache et al., 2011). It encompasses several tissue types, including ductal carcinoma in situ and lobular carcinoma in situ (DCIS/LCIS), non-cancerous (Healthy) tissue, invasive ductal carcinoma (IDC), and tumor edge areas with relatively low malignancy (Tumor edge). This dataset covers 20 spatial regions and measures the expression of 36,601 genes on 3798 spots. The third dataset contains mouse anterior brain (MAB) tissue with spatial annotations for 2695 spots on 52 regions. Each sample in MAB captures the expression levels of 32,285 genes.

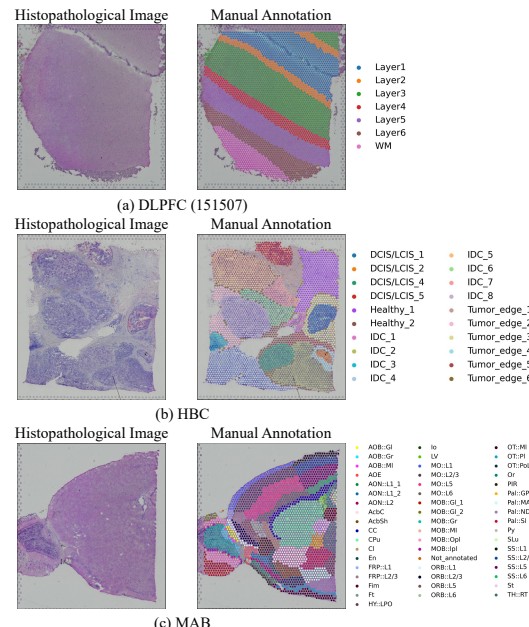

Figure 17: Histopathological images and their corresponding manual annotations images.

**Baselines.** We select nine representative methods for comparison, including SCANPY, SpaGCN, DeepST, STAGATE, GraphST, Smoother, SCGDL, stLearn and DUSTED, described as follows.

- **SCANPY (Wolf et al., 2018)** provides an toolkit for analyzing single-cell gene expression data. It includes functions for preprocessing, dimensionality reduction, visualization, and clustering, offering researchers a powerful toolset for managing single-cell RNA sequencing data.

- **SpaGCN (Hu et al., 2021)** utilizes GCNs effectively to combine gene expression data, spatial coordinates, and histological images, thereby facilitating the identification of spatial domains.

- **DeepST (Xu et al., 2022)** identifies spatial domains in SRT data with high accuracy, supporting detailed analysis of cancer tissues, data integration from diverse sources, and extension to other forms of spatial muti-omics data.

- **STAGATE (Dong & Zhang, 2022b)** decodes spatial domains using spatial graph attention networks, where attention mechanisms model spatial dependencies between spots, maintaining continuous spatial expression gradients in transcriptomic data.

- **GraphST (Long et al., 2023)** combines GNNs with self-supervised contrastive learning to capture distinct representations and spot-specific information, helping to reveal different cell types.

- **Smoother (Su et al., 2023)** integrates positional data into non-spatial models using modular priors, enabling precise data imputation, cell-type deconvolution, and dimensionality reduction.

- **SCGDL (Liu et al., 2023)** integrates deep graph infomax (DGI) with a residual gated graph convolutional network for the identification of spatial domains.

- **stLearn (Pham et al., 2023)** merges morphological features from tissue images with gene expression data from adjacent spots to efficiently cluster similar spatial domains within tissues.

- **DUSTED (Zhu et al., 2025)** proposes a denoising framework that integrates a graph autoencoder architecture with complementary gene channel attention and graph attention mechanisms to delineate spatial domains in the presence of biological noise.

**Implement Details.** To preprocess datasets, we initially remove genes exhibiting very low expression or variance. We then retain the top 2,000 genes exhibiting the highest variance (3,000 for the DLPFC 151507 dataset). Using this selection, we construct an undirected spatial graph by calculating the Euclidean distance within a fixed radius $r$ (set to 550 $\mu m$ in our experiments), which helps capture the spatial relationships. For $k$-nearest neighbors in feature graph, we set $k = 14$ for the first two datasets, and $k = 15$ for the third dataset. Our stUAI is implemented using the PyTorch v2.4 framework and trained on a single NVIDIA RTX 4090 GPU with 24GB of memory. For optimization, we employ the AdamW optimizer, with a learning rate of 1e-3 for all datasets.

