# OpenReview forum: "stUAI: Uncertainty-Aware Clustering of Spatially Resolved Transcriptomics Data"
_ICLR.cc/2026/Conference — Submitted to ICLR 2026_

### Official Review · Reviewer_2U3y · 2025-10-29

**Soundness:** 2
**Presentation:** 2
**Contribution:** 2
**Rating:** 4
**Confidence:** 4

**Summary:**

This paper introduces probabilistic embedding approach for spatial transcriptomics data. They propose intra-view contrastive learning and cross-view information alignment for embedding, and evaluated in spatial transcripotmics data.

**Strengths:**

This work first employ probabilistic embedding for spatial transcriptomics.

**Weaknesses:**

- Limited Novelty: Both probabilistic embedding and multiple modality are existing methodologies. This paper has merely applied them to the field of spatial transcriptomics.

- Source of Uncertainty: It is unclear where the uncertainty originates. In this paper, the distribution is primarily learned with a focus on the intra-view data. What does the variance in this context signify? For example, if variance were learned in the cross-view, one could attribute it to the difference between the two modalities. However, what causes this variance in the intra-view setting?

- Utility of Uncertainty: How can the uncertainty be utilized? The case in Section 3.5 is a post-hoc analysis. What practical action can be taken if the model determines the uncertainty is high?

- Alternative Divergences (Section 2.3): What would happen to the model if it were trained using KL Divergence or JS Divergence instead of the current formulation in Section 2.3?

- Benefit to Uni-modal Data: Can Spatial + Single Cell multimodal training also benefit single cell unimodal data?

- Missing Reference: The following reference is missing: Global Context-aware Representation Learning for Spatially Resolved Transcriptomics, ICML 2025.

**Questions:**

See Weakness Section

---

> ### Author Response · Authors · 2025-11-21
>
> We are truly grateful for the time you have taken to review our paper and your insightful review. Here we address your comments in the following, where Q denotes the comment of the reviewer and R denotes our response.
>
> > Q1. Limited Novelty: Both probabilistic embedding and multiple modality are existing methodologies. This paper has merely applied them to the field of spatial transcriptomics.
>
> R1. Thanks for your valuable comment! While probabilistic embeddings and multimodal learning are indeed established concepts, they remain highly active research directions, and numerous works continue to propose new models within these paradigms. The existence of these concepts does not imply a lack of novelty; rather, innovations arise from how these ideas are adapted, reformulated, and optimized for specific data structures and domain challenges.
>
> Our work is, to the best of our knowledge, the first to introduce probabilistic embedding learning into ST analysis, particularly for spot-level distributional representations. Moreover, beyond simply applying existing ideas, we make several methodological innovations tailored to ST data: **(1)** a cluster-guided contrastive learning framework that exploits distributional semantics to enhance representation stability and discriminability, **(2)** an OT-based uncertainty alignment module that aligns probabilistic semantics across views in a principled manner. These components form a unified self-supervised framework that enables stable, discriminative, and semantically meaningful representations, ultimately leading to substantial improvements in ST analysis. Thus, the contribution of our stUAI is not a straightforward application of existing methods, but a novel modeling perspective and algorithmic design tailored to ST.
>
> > Q2. Source of Uncertainty: It is unclear where the uncertainty originates. In this paper, the distribution is primarily learned with a focus on the intra-view data. What does the variance in this context signify? For example, if variance were learned in the cross-view, one could attribute it to the difference between the two modalities. However, what causes this variance in the intra-view setting?
>
> R2. Thanks for your valuable problem! We indeed learn intra-view distributional representations. The variance of these distributions primarily characterizes each spot’s semantic uncertainty within that view, also referred to as data-inherent uncertainty. This uncertainty reflects the intrinsic ambiguity or fuzziness caused by noise inherent to the data itself, rather than arising from the model’s decision process, and it is not used to measure discrepancies across different views.
>
> Within each view, data-inherent uncertainty originates from both the input gene expression matrix and the constructed graph structure. In particular, ST data are typically highly sparse, overly dispersed, and noisy. RNA contamination, capture efficiency, and sequencing depth would introduce fluctuations in the observed expressions. Moreover, the constructed graph structure may not perfectly reflect the true underlying topology. As a result, such factors lead to semantic ambiguity that cannot be adequately captured using a single point embedding. Therefore, for each view, we model a distributional representation for each spot, capturing the intrinsic ambiguity of each spot’s semantic representation caused by data noise, spatial uncertainty, and measurement variability, rather than uncertainty stemming from the model’s decision process.

---

> > ### Author Response · Authors · 2025-11-21
> >
> > > Q3. Utility of Uncertainty: How can the uncertainty be utilized? The case in Section 3.5 is a post-hoc analysis. What practical action can be taken if the model determines the uncertainty is high?
> >
> > R3. Thanks for your valuable problem! Figure 3.5 offers an intuitive validation of the plausibility of the learned uncertainty, showing from the perspective of dropout events that our distributional representation indeed captures the inherent uncertainty of the data. In our stUAI framework, we utilize the learned uncertainty for representation fusion, where we adopt an inverse-variance weighting strategy to downweight samples with high uncertainty, thereby reducing the negative influence of uncertainty and achieving reliable and stable representations. Consequently, when two views are fused, spots associated with higher uncertainty in one view receive smaller weights, mitigating the impact of highly uncertain components on the final domain identification result.
> >
> > In addition, one could consider removing spots that exhibit abnormally high uncertainty in both views before performing clustering, thereby reducing the risk of forming spurious clusters. Uncertainty can also be used to weight each spot’s expression values during differential expression analysis, improving the reliability of detected differential expression genes. Moreover, uncertainty visualization, similar to Figure 6, can assist pathologists or biologists in assessing which regions yield trustworthy analytical results and can be overlaid with H&E images for visual validation. These directions provide promising opportunities for further investigation, and we will explore them in future work.
> >
> > **Please refer to the blue-highlighted content in Appendix H in the revised manuscript**.
> >
> > > Q4. Alternative Divergences (Section 2.3): What would happen to the model if it were trained using KL Divergence or JS Divergence instead of the current formulation in Section 2.3?
> >
> > R4. Thanks for your valuable suggestion! We replace our OT with KL and JS divergences for comparison. The results in the table below show that our OT still outperforms the other two ways. Compared with KL and JS divergences, OT better handles distributions with differing supports and aligns complex high-dimensional distributions. In addition, the distributional representations are Gaussian, which can be fully characterized by their first- and second-order moments.
> >
> > | **Dataset** | **Metric** | **KL** | **JS** |**OT (Ours)** |
> > |-------------|------------|--------|--------|------------------|
> > | **HBC**     | ARI        | 0.60   | 0.61   |**0.65**       |
> > |             | NMI        | 0.66   | 0.65   | **0.69**       |
> > | **MAB**     | ARI        | 0.46   | 0.47   |**0.49**       |
> > |             | NMI        | 0.67   | 0.69   |**0.71**       |
> >
> >
> > > Q5. Benefit to Uni-modal Data: Can Spatial + Single Cell multimodal training also benefit single cell unimodal data?
> >
> > R5. Thanks for your valuable problem! Yes, our Spatial + Single Cell multimodal training framework can indeed benefit the analysis of single-cell unimodal data. This can be achieved through two primary strategies that translate unimodal data into a form compatible with our multimodal learning paradigm: (1) **Using a KNN graph as a proxy for spatial context:** A common practice in single-cell analysis is to construct a K-nearest neighbor (KNN) graph from the gene expression matrix to capture cellular relationships. In our framework, this graph can serve as a direct surrogate for the physical spatial network. The model, trained to reconcile gene expression with spatial organization, can thus be applied by using the expression profile as one modality and the KNN graph's adjacency information as the second, structural modality. This allows the model's spatially-aware learning to enhance the analysis of standard scRNA-seq data. (2) **Creating augmented views for pseudo-multimodal training:** Separately, our training methodology is directly applicable to feature augmentation techniques commonly used in single-cell contrastive learning. Here, the original feature matrix can be used to generate multiple augmented views (e.g., via masking, adding noise, or subsampling). These views can be treated as distinct "pseudo-modalities" and fed into our multimodal training framework. The model then learns to extract robust, integrative representations by identifying consistent biological signals across these artificially generated views, thereby significantly improving downstream tasks.
> >
> > In both scenarios, the core advantage lies in the transfer of our model's ability to integrate cross-modal information. This demonstrates that the insights gained from true multimodal training can be generalized to benefit the vast universe of existing unimodal single-cell datasets.

---

> > > ### Author Response · Authors · 2025-11-21
> > >
> > > > Q6. Missing Reference: The following reference is missing: Global Context-aware Representation Learning for Spatially Resolved Transcriptomics, ICML 2025.
> > >
> > > R6. Thanks for your valuable suggestion! We have now included and discussed this latest work in both the Introduction (Section 1) and Related Work (Appendix B.1) sections.
> > >
> > > **Please refer to the blue-highlighted content in Section 1 and Appendix B.1 in the revised manuscript**.
> > >
> > > In light of these responses, we hope we have addressed your concerns, and hope you could consider raising your score. If there are any additional notable points of concern that we have not yet addressed, please do not hesitate to share them, and we will promptly attend to those points.

---

### Official Review · Reviewer_Rfbg · 2025-11-01

**Soundness:** 3
**Presentation:** 3
**Contribution:** 2
**Rating:** 4
**Confidence:** 5

**Summary:**

The authors note that most existing methods struggle with effective denoising and suffer from model uncertainty, which reflects inherent semantic ambiguity and limited knowledge sharing between spatial and expression modalities. To address these challenges, they propose stUAI, an Uncertainty-Aware Integration framework for clustering spatially resolved transcriptomics data. The framework quantifies uncertainty in both modalities and performs cross-view distributional representation alignment using an optimal transport algorithm. In addition, the authors introduce Cluster-Guided Intra-View Contrastive Learning (CGCL), which defines confidence based on the distance to the cluster center and treats a pair as positive if both samples are confident and belong to the same cluster. The fusion of the two modalities is then carried out based on the learned uncertainty and optimized through reconstruction and spatial regularization objectives. The proposed method is evaluated on the HBC, DLPFC, and MBA tissue datasets.

**Strengths:**

- The authors propose a well-founded method that effectively addresses an important challenge in spatially resolved transcriptomics—quantifying uncertainty and learning a fused representation from two distinct modalities.

- The presentation of the work is clear and easy to follow.

**Weaknesses:**

- The dataset used in this study is limited, which is a major weakness and undermines the sufficiency of the experimental validation.

- The ablation study on the fusion strategy presented in Table 3 only evaluates a naïve averaging approach; additional comparisons with more advanced fusion strategies would strengthen the analysis.

- Although the authors attempt to demonstrate the reliability of the learned uncertainty in Section 3.5, the provided evidence is insufficient to fully support their claims.

**Questions:**

- Please provide spatial domain identification results on additional datasets, such as MERFISH, to further validate the generalizability of the proposed method.

- Include a comparison with attention-based fusion approaches to better demonstrate the effectiveness of the proposed fusion strategy.

- The current validation of the learned uncertainty is insufficient. The authors are encouraged to test whether spots with lower uncertainty are more likely to be well-clustered, for example, by evaluating this relationship using cluster accuracy (CA) metrics.

- Please provide more detailed information about the preprocessing steps applied to the DLPFC dataset.

---

> ### Author Response · Authors · 2025-11-21
>
> We are truly grateful for the time you have taken to review our paper and your insightful review. Here we address your comments in the following, where Q denotes the comment of the reviewer and R denotes our response.
>
> > Q1. Please provide spatial domain identification results on additional datasets, such as MERFISH, to further validate the generalizability of the proposed method.
>
> R1. Thanks for your valuable suggestion! Thanks for your valuable suggestion! We additionally include a new dataset, the mouse olfactory bulb, which was sequenced using the Stereo-seq platform, and contains 19109 spots and 14376 genes across seven tissue layers, i.e., the olfactory nerve layer (ONL), glomerular layer (GL), external plexiform layer (EPL), mitral cell layer (MCL), internal plexiform layer (IPL), granule cell layer (GCL), and rostral migratory stream (RMS). We evaluate the performance of our stUAI, GraphST, and DUSTED on domain identification using this dataset, and report the corresponding Moran’s I scores in the table below.
>
> |  | GraphST | DUSTED | stUAI
> ---|-----|-----|----|
> Moran’s I | 0.59 | 0.28 | 0.67 |
>
> It can be seen that our stUAI yields clustering results with stronger spatial autocorrelation on this dataset. It further validates the effectiveness and generalizability of our method.
>
>
> > Q2. Include a comparison with attention-based fusion approaches to better demonstrate the effectiveness of the proposed fusion strategy.
>
> R2. Thanks for your valuable suggestion! Here we compare the results of the attention-based fusion strategy with our uncertainty-based fusion approach, as shown in the table below.
>
> | **Dataset** | **Metric** | **Attention** | **Uncertainty (Ours)** |
> |-------------|------------|--------|------------------|
> | **DLPFC**   | ARI        | 0.78   | **0.84**       |
> |  **(151671)**  | NMI        | 0.71   | **0.78**       |
> | **HBC**     | ARI        | 0.62   | **0.65**       |
> |             | NMI        | 0.68   | **0.69**       |
> | **MAB**     | ARI        | 0.47   | **0.49**       |
> |             | NMI        | 0.69   | **0.71**       |
>
> As observed from the table above, though attention-based fusion strategy effectively integrate feature modality and spatial modality information, its performance is still inferior to our uncertainty-based fusion strategy, which is because our strategy captures uncertainties arising from technical or intrinsic data factors, mitigating their negative impact to ensure that representation components with higher uncertainty contribute less to the fused representation.
>
> **Please refer to the blue-highlighted content in Table 3 and the second part of Section 3.3 in the revised manuscript**.
>
>
> > Q3. The current validation of the learned uncertainty is insufficient. The authors are encouraged to test whether spots with lower uncertainty are more likely to be well-clustered, for example, by evaluating this relationship using cluster accuracy (CA) metrics.
>
> R3. Thanks for your valuable suggestion! In our method, the uncertainty we measure arises from the learned distributional representations in Eq. (1), which provide a fuzzy depiction of semantics and reflect spot-level ambiguity induced by artificial or technical noise. Therefore, the uncertainty we consider characterizes the semantic fuzziness inherent to each spot, which is primarily caused by data noise rather than model decision-making.
> These distributional representations are used to construct our self-supervised losses, i.e., the cluster-guided contrastive loss and the OT-based uncertainty alignment loss. The former encourages the model to leverage semantically meaningful distributions, so that the contrastive loss can more appropriately enforce similarity among positive pairs and discriminability among negative pairs, yielding more stable and higher-quality embeddings. The latter aims to better align the semantics of the two views, which are represented as probability distributions.
> In addition, during representation fusion, we adopt an inverse-variance weighting strategy to downweight samples with high uncertainty, thereby reducing their influence on the fused representation. Consequently, the magnitude of uncertainty may be related to clustering performance; for instance, if certain spots exhibit much higher uncertainty than others in both views, the final clustering performance for these spots may indeed be poor, as their distributional representations are largely dominated by noise. However, in other cases, due to the inverse-variance weighting, the intra-view uncertainty does not necessarily have a strong correlation with the final clustering performance. If the focus is on epistemic uncertainty, i.e., uncertainty in the model’s predictive performance, its relationship to clustering quality would be stronger. This is not the type of uncertainty addressed in our work.

---

> > ### Author Response · Authors · 2025-11-21
> >
> > > Q4. Please provide more detailed information about the preprocessing steps applied to the DLPFC dataset.
> >
> > R4. Thanks for your valuable suggestion! In the preprocessing of the DLPFC dataset, we use the *scanpy* package in Python. We first remove genes that are not expressed in at least 100 cells using the `sc.pp.filter_genes` function. Then, following the Seurat v3 procedure, we select the top 2000 highly variable genes using `sc.pp.highly_variable_genes`. Next, we perform normalization by scaling each spot’s gene expression values to a total count of 10000, thereby reducing the influence of sequencing depth. Finally, we standardize the data using `sc.pp.scale`, capping each gene’s expression value at 10 to mitigate the effect of extreme values on downstream analysis.
> >
> > In light of these responses, we hope we have addressed your concerns, and hope you could consider raising your score. If there are any additional notable points of concern that we have not yet addressed, please do not hesitate to share them, and we will promptly attend to those points.

---

### Official Review · Reviewer_XHFa · 2025-11-01

**Soundness:** 3
**Presentation:** 3
**Contribution:** 2
**Rating:** 4
**Confidence:** 4

**Summary:**

The paper introduces stUAI, a novel framework for representation learning in Spatially Resolved Transcriptomics (SRT) data. It aims to overcome (1) the spasity and over-dispersion in SRT data, (2) sampling bias in contrative learning, and (3) shallow guidance and knowledge sharing between modality-specific representation (spatial and gene expression).

To address these issue, stUAI introduces three components:

1. Cluster-guided Intra-view Contrastive Learning: Creates positive pairs without augmentation using distributional representations and the reparameterization trick. Contrastive learning is then performed using reliable positive pairs, which must satisfy two conditions: (1) both spots are high-confidence (close to their cluster center), and (2) both spots are predicted to be in the same cluster.

2. Cross-view Distributional Representation Alignment: Employs Optimal Transport (OT) to ensure that the two different views—spatial (position) information and feature (gene) information—produce consistent predictions.

3. Representation Fusion & ZINB Reconstruction: Fuses spatial and gene information by weighting them based on uncertainty (derived from the inverse of their variances), normalizing these weights via softmax, and then performing a ZINB-based reconstruction.

To verify its effectiveness, the model was evaluated on clustering, DEG analysis, trajectory inference, gene imputation, and gene pathway enrichment, in addition to uncertainty analysis, demonstrating strong performance.

**Strengths:**

* The paper clearly explains its key motivations. Additionally, it proposes a methodology that aligns well with these motivations.
* The method appropriately incorporates uncertainty and validates this inclusion through experiments.
* The paper is well-written and easy-to-follow.

**Weaknesses:**

* The time complexity is a concern, as the model appears to perform k-means clustering and Optimal Transport (OT) in every epoch.
* The justification for the architecture is not fully convincing. The model uses CGCL on each modality before fusing them and uses OT for alignment. Why not apply CGCL directly to the final fused representation? Alternatively, why not integrate the graphs at the input level rather than aligning the representations via OT?
* The motivation should be more focused on the target domain. The ZINB loss is a common component in single-cell analysis and not a novel motivation in itself. The paper motivates its contrastive learning approach by citing general limitations (e.g., sampling bias). A stronger motivation would be to directly link this to specific SRT challenges. For instance, the high sparsity and over-dispersion in SRT data make traditional augmentation strategies ineffective for generating reliable positive/negative pairs.

**Questions:**

* What is the time complexity of stUAI compared to the baselines?
* Performance in unsupervised settings can often be unstable.
    * How many times were the experiments run (e.g., with different random seeds)?
    * Please provide statistical tests (e.g., p-values) to demonstrate the robustness of stUAI over baselines.
    * In conventional approaches, preprocessing steps and hyperparameters are typically fixed due to the lack of a validation set. Is there a specific reason for using different HVG counts (2000 vs. 3000) and k-nearest neighbor counts (14 vs. 15)2? Could you provide results where these are fixed to the same values across datasets?
* In the ablation study (Table 2), does "stUAI w/o $\mathcal{L}_{ZINB_{reg}}$" remove the reconstruction loss entirely? What is the performance if the ZINB loss is replaced with a standard reconstruction loss, such as MSE?
* In the fusion strategy comparison (Table 3), how does the proposed uncertainty-aware fusion ($M_4$) compare to a more common approach using learnable gating parameters?
* The sensitivity analysis (Figure 4) explores the $k$ for the feature graph neighbors. However, since k-means clustering is used during training, what is the model's sensitivity to the number of clusters ($K$) used in k-means?
* How are the "uncertainty scores" calculated in Section 3.5 (Figure 6b)?
* The pseudotime trajectory inference results (Figure 7) appear unusual. Typically, the trajectory in the DLPFC dataset is expected to originate from White Matter (WM) and proceed toward Layer 1. It would also be beneficial to compare this trajectory with those generated by other baselines, as in [1].
* For the gene imputation task (Figure 8), how does the model performance vary with different random mask ratios 5, as analyzed in [2]?
---
[1] Identifying multicellular spatiotemporal organization of cells with SpaceFlow. Nature communications. 2022.

[2] Global Context-aware Representation Learning for Spatially Resolved Transcriptomics. ICML. 2025.

---

> ### Author Response · Authors · 2025-11-21
>
> We are truly grateful for the time you have taken to review our paper and your insightful review. Here we address your comments in the following, where Q denotes the comment of the reviewer and R denotes our response.
>
> > Q1. What is the time complexity of stUAI compared to the baselines?
>
> R1. Thanks for your valuable suggestion! Here we compare the per-iteration running time of two latest methods across two datasets, as shown in the table below.
>
> | **Dataset** | **stLearn** | **DUSTED** | **stUAI (Ours)** |
> |-------------|------------|--------|------------------|
> | **HBC**     | >1s   | 42.25 ms   |    120.63 ms   |
> | **MAB**     | >1s   | 35.10 ms   |    82.17 ms   |
>
> The results show that our method overall maintains a high computational efficiency, positioning itself between two state-of-the-art baselines and significantly lower than stLearn’s runtime. Compared with several baselines, our approach also delivers substantially better performance than all competing methods. This observation further confirms that our method offers superior performance gains while preserving high efficiency.
>
> > Q2. The justification for the architecture is not fully convincing. The model uses CGCL on each modality before fusing them and uses OT for alignment. Why not apply CGCL directly to the final fused representation? Alternatively, why not integrate the graphs at the input level rather than aligning the representations via OT?
>
> R2. Thanks for your valuable comment! In our stUAI, we consider applying CGCL to the distributional representations learned from the spatial and feature views separately. Compared with applying CGCL directly on the final fused representation, our method helps improve the representation quality of each view's encoder, enabling each view to learn stable and discriminative embeddings. It also facilitates subsequent alignment: without view-wise CGCL, the view-specific differences may be suppressed during alignment, leading to information loss. Moreover, the alignment loss only encourages the two views to be close to each other and cannot determine which view is clean. View-wise CGCL ensures that the embeddings used for alignment are already high-quality semantic representations, so the alignment loss operates on two meaningful spaces rather than aligning noise, thereby reducing the risk of noise propagation through the alignment process while also improving the quality of representations used for fusion and preserving more useful information. Finally, the entire process follows a "optimize first, then fuse" pattern, where each component performs its dedicated role, avoiding complex gradient interactions and mutual interference, thus making the entire training process more stable.
>
> Furthermore, compared to directly fusing at the graph data level, representation-level alignment first learns node embeddings independently for spatial and feature graphs before performing alignment in the embedding space, which offers several advantages: it preserves modality-specific features by allowing each view to focus on capturing its own structural and attribute information without feature conflicts caused by view differences in the fused graph; it reduces noise interference by only optimizing in the embedding space, minimizing the risk of direct noise transfer between modalities; and it provides flexibility and scalability through independent encoding for each modality that facilitates incremental updates and eliminates the need to reconstruct the entire large graph when adding new modalities, thus avoiding computational explosion resulting from an overly large or dense fused graph.

---

> ### Author Response · Authors · 2025-11-21
>
> > Q3. The motivation should be more focused on the target domain. The ZINB loss is a common component in single-cell analysis and not a novel motivation in itself. The paper motivates its contrastive learning approach by citing general limitations (e.g., sampling bias). A stronger motivation would be to directly link this to specific ST challenges. For instance, the high sparsity and over-dispersion in ST data make traditional augmentation strategies ineffective for generating reliable positive/negative pairs.
>
> R3. Thanks for your valuable comment! Traditional contrastive learning generates augmentations by randomly perturbing features or adding/removing edges, and uses only each sample’s own augmentation as the positive pair. In highly sparse and over-dispersed ST data, such augmentations are prone to produce prominent outlier expressions or noise, causing the contrastive learning to be dominated by extreme values, which exacerbates the instability and semantic bias of the embeddings. Moreover, using only self-augmentations as positives fails to leverage the shared patterns among similar cells, further reducing the robustness of embeddings to sparse and over-dispersed data. Therefore, we aim to establish augmentations through sampling based on distributional representations and utilize high-confidence clustering information to select reliable positive pairs, thereby promoting more meaningful contrastive learning.
>
> **Please refer to the blue-highlighted content at the beginning of Section 2.2 in the revised manuscript.**
>
> > Q4. Performance in unsupervised settings can often be unstable.
> How many times were the experiments run (e.g., with different random seeds)?
> Please provide statistical tests (e.g., p-values) to demonstrate the robustness of stUAI over baselines.
> In conventional approaches, preprocessing steps and hyperparameters are typically fixed due to the lack of a validation set. Is there a specific reason for using different HVG counts (2000 vs. 3000) and k-nearest neighbor counts (14 vs. 15)? Could you provide results where these are fixed to the same values across datasets?
>
> R4. Thanks for your valuable problems!
> The experimental results are averaged over five runs with different random seeds.
> To validate the significance of our superior result, we perform rank-sum tests between competitive baselines and stUAI over 12 slices of DLPFC, with p-values shown below:
>
> |             |**SpaGCN**   |**DeepST**   |**STAGATE**    |**GraphST**    |**DUSTED**
> |     -       |-        |-          |-          |-        |-
> |**p-value (ARI)**|6e-05    |0.0019   |0.0003     |0.0347     |0.0022
> |**p-value (NMI)**|6e-05    |0.0191   |0.0040     |0.0638     |0.0876
>
> The results demonstrate that at the 0.05 significance level, 8 out of 10 values are statistically significant (p < 0.05), providing substantial evidence that our method achieves statistically superior performance compared to the competitive baselines.
>
> Moreover, as shown below, after unifying the experimental settings (HVG=2000, k=14), our method stUAI still achieves optimal performance compared to other baselines. We sincerely appreciate your valuable feedback on this inconsistency and will ensure all experiments adhere to this unified configuration in the revised version.
>
> |              |**ARI** |**NMI**
> |-             |-   |-
> |**DLPFC (151507)**|0.66|0.71
> |**MAB**                 |0.46|0.72
>
> > Q5. In the ablation study (Table 2), does "stUAI w/o $\mathcal{L}{ZINB{reg}}$" remove the reconstruction loss entirely? What is the performance if the ZINB loss is replaced with a standard reconstruction loss, such as MSE?
>
> R5. Yes, we remove the reconstruction loss entirely. Here, we replace ZINB-based reconstruction with MSE reconstruction, the results are presented as below.
>
> | **Dataset** | **Metric** | **MSE** | **ZINB (Ours)** |
> |-------------|------------|--------|------------------|
> | **DLPFC**   | ARI        | 0.80   | **0.84**       |
> |   **(151671)**  | NMI        | 0.75   | **0.78**       |
> | **HBC**     | ARI        | 0.60   | **0.65**       |
> |             | NMI        | 0.64   | **0.69**       |
> | **MAB**     | ARI        | 0.44   | **0.49**       |
> |             | NMI        | 0.67   | **0.71**       |
>
> As observed from the table above, comparing the MSE reconstruction with our ZINB-based reconstruction reveals the inferior performance of MSE relative to our ZINB. This performance gap stems from MSE's inability to adequately capture the sparse and over-dispersion characteristics of single-cell sequencing data, whereas our ZINB effectively models these key data properties.
>
> **Please refer to the blue-highlighted content in Table 6, Appendix E in the revised manuscript**.

---

> ### Author Response · Authors · 2025-11-21
>
> > Q6.  In the fusion strategy comparison (Table 3), how does the proposed uncertainty-aware fusion ($M_4$) compare to a more common approach using learnable gating parameters?
>
> R6.  Thanks for your valuable suggestion! Here we compare the results of the attention-based fusion strategy with our uncertainty-based fusion approach, as shown in the table below.
>
> | **Dataset** | **Metric** | **Attention** | **Uncertainty (Ours)** |
> |-------------|------------|--------|------------------|
> | **DLPFC**   | ARI        | 0.78   | **0.84**       |
> |  **(151671)** | NMI        | 0.71   | **0.78**       |
> | **HBC**     | ARI        | 0.62   | **0.65**       |
> |             | NMI        | 0.68   | **0.69**       |
> | **MAB**     | ARI        | 0.47   | **0.49**       |
> |             | NMI        | 0.69   | **0.71**       |
>
> As observed from the table above, though attention-based fusion strategy effectively integrate feature modality and spatial modality information, its performance is still inferior to our uncertainty-based fusion strategy, which is because our strategy captures uncertainties arising from technical or intrinsic data factors, mitigating their negative impact to ensure that representation components with higher uncertainty contribute less to the fused representation.
>
> **Please refer to the blue-highlighted content in Table 3 and the second part of Section 3.3 in the revised manuscript**.
>
> > Q7. The sensitivity analysis (Figure 4) explores the $k$ for the feature graph neighbors. However, since k-means clustering is used during training, what is the model's sensitivity to the number of clusters ($K$) used in k-means?
>
> R7. Thanks for your valuable suggestion! Here, we experiment with different numbers of clusters ($K$) across three datasets, and the results are shown in the table below.
>
> DLPFC (151507):
>
> |$K$           |5      |6      |7     |8
> |     -        |-        |-      |-      |-
> |ARI           |0.66   |0.67   |**0.70**  |0.69
> |NMI          |0.73   |0.73   |**0.74**   |0.71
>
> HBC:
>
> |$K$           |16       |18     |20     |22
> |     -        |-        |-      |-      |-
> |ARI           |0.61     |0.64   |**0.65**   |0.62
> |NMI           |0.63     |0.66   |**0.69**   |0.66
>
> MAB:
>
> |$K$           |48       |50     |52     |54
> |     -        |-        |-      |-      |-
> |ARI           |0.48     |0.49   |**0.49**   |0.47
> |NMI           |0.67     |0.70   |**0.71**   |0.68
>
> It can be observed that when $K$ is set to a value different from the actual number of classes, the performance experiences a slight decline. However, overall, when the set number of clusters is close to the true number of classes, the performance does not show significant degradation.
>
> > Q8. How are the "uncertainty scores" calculated in Section 3.5 (Figure 6b)?
>
> R8. Thanks for your valuable problem! The uncertainty visualization in Figure 6b is computed from the distributional representations learned in the spatial view. We present the spatial view here because it incorporates both spatial coordinates and gene expression information. Specifically, after training, we obtain a distributional representation for each spot in the spatial view. We extract the variance vector of each distribution and take its average as the uncertainty score for that spot, which is then visualized at the corresponding spatial coordinates to produce Figure 6b.
>
> **Please refer to the blue-highlighted content in Section 3.5 in the revised manuscript**.
>
> > Q9. The pseudotime trajectory inference results (Figure 7) appear unusual. Typically, the trajectory in the DLPFC dataset is expected to originate from White Matter (WM) and proceed toward Layer 1. It would also be beneficial to compare this trajectory with those generated by other baselines, as in [1].
>
> R9. Thanks for your insightful comments and suggestions. The trajectory plot in the previous version indeed contained some inaccuracies, mainly due to an incorrect selection of the root cell in the trajectory inference algorithm (we used the DPT algorithm, which is based on diffusion-like random walks). In the revised version, we correct this issue and further compare our method with SCANPY and DUSTED. Our stUAI produces clearer hierarchical trajectory structures and smoother color transitions. The trajectory inferred from stUAI also shows a stronger alignment with the manual annotation compared with SCANPY and DUSTED, and it originates from WM and progresses toward Layer 3, which is consistent with widely recognized biological facts and observations.
>
> **Please refer to the blue-highlighted content in Section 3.6 and Figure 7 in the revised manuscript**.

---

> > ### Author Response · Authors · 2025-11-21
> >
> > > Q10. For the gene imputation task (Figure 8), how does the model performance vary with different random mask ratios, as analyzed in [2]?
> >
> > R10. Thanks for your valuable suggestion! Here, we select two genes, MBP from DLPFC dataset and COL6A1 from HBC dataset, to conduct the gene imputation task. We randomly mask varying proportions of gene expression and utilize different algorithms to perform imputation. Using Pearson Correlation Coefficient (PCC) as the evaluation metric, we compare the DUSTED method with our proposed approach stUAI, with the quantitative results summarized in the table below.
> >
> > MBP (PCC)
> >
> > |mask ratios  |10%      |20%    |30%    |40%   |50%
> > |     -       |-        |-      |-      |-     |-
> > |DUSTED    |0.39     |0.38   |0.42   |0.30  |0.33
> > |stUAI (Ours)  |0.75     |0.78   |0.72   |0.74  |0.71
> >
> > COL6A1 (PCC)
> >
> > |mask ratios  |10%      |20%    |30%    |40%   |50%
> > |     -       |-        |-      |-      |-     |-
> > |DUSTED |0.45     |0.44   |0.41   |0.42  |0.41
> > | stUAI (Ours) |0.52     |0.52   |0.50   |0.51  |0.49
> >
> > The results clearly demonstrate that for the MBP gene, our method stUAI achieves substantially better imputation accuracy across all masking ratios compared to DUSTED. Similarly, for the COL6A1 gene, our approach consistently outperforms the method DUSTED. These findings further confirm the superior performance of our method in gene imputation tasks.
> >
> > In light of these responses, we hope we have addressed your concerns, and hope you could consider raising your score. If there are any additional notable points of concern that we have not yet addressed, please do not hesitate to share them, and we will promptly attend to those points.

---

> > > ### Comment · Reviewer_XHFa · 2025-11-27
> > >
> > > I acknowledge the authors' efforts in the rebuttal, which has addressed most of my previous concerns. However, regarding the experimental results, I have one remaining suggestion to further strengthen the paper's robustness.
> > >
> > > While the main performance comparison (Table 1) covers all slices, most other experiments or those presented in the rebuttal focus on results **for a single specific slice, and the selected slice varies across experiments**. Since there are variations among the 12 DLPFC slices, I strongly recommend adding the averaged results or detailed results for all 12 slices to the Appendix to demonstrate that the effectiveness of the proposed modules is not limited to a specific slice.

---

> > > > ### Author Response · Authors · 2025-11-28
> > > >
> > > > Thanks for your valuable suggestion! For the main experimental results, Figure 3 in the previous version already reports the average performance across all slices of the DLPFC dataset, where our method achieves the best results on both metrics. In addition, to more comprehensively demonstrate the robustness of our model across different slices, we provide the full and average results on all 12 DLPFC slices for the ablation studies, fusion strategies, and sensitivity analyses, as shown in Figures 7, 8, and 10 in Appendix E. For the ablation studies and fusion strategies, our method produces conclusions largely consistent with those from the single-slice analysis on most slices, with only minor variations on a few slices. Overall, based on the average performance across slices, our stUAI consistently outperforms all variants, further demonstrating its robustness and superior generalization ability across different slices. Meanwhile, the sensitivity analysis across different slices further shows that our hyperparameter settings generalize well across slices.
> > > >
> > > > **Please refer to the blue-highlighted content in Figure 3 (Section 3.2) and Tables 7, 8, and Figure 10 in Appendix E in the revised manuscript**.
> > > >
> > > > We hope we have addressed your concerns, and we appreciate your time and valuable comments on our manuscript!

---

### Official Review · Reviewer_bht3 · 2025-11-01

**Soundness:** 2
**Presentation:** 2
**Contribution:** 2
**Rating:** 4
**Confidence:** 4

**Summary:**

This paper works on the clustering of spatial transcriptomics sites. In addition to fusing the spatial coordinates and the gene readouts, the authors add an uncertainty quantification part where embeddings from each modality are passed through or trains a parametrized gaussian distribution which introduces four (mean and std x 2) heads to represent each set of embeddings as a gaussian distribution. With such modeling of each modality’s embeddings as a distribution, the authors also propose to guide the learning of such distribution (which will backprop to embeddings representations) using pseudo-label (clustered) guided contrastive learning. In order to make sure each modality’s distribution truly associates with the other, the authors also apply an optimal transport method to align two distributions. Then the two embeddings are fused together and guided by ZINB reconstruction with a spatial regularizer.
The authors perform experiments on three popular datasets and report the ARI and NMI for different spatial clustering methods.

**Strengths:**

1. This paper incorporates uncertainty or distribution modelling in the method and also utilized the newly introduced data distribution network to perform contrastive learning supervision and optimal transport to force/further fuse the two modalities.
2. The authors compared an abundance of prior methods that conduct clustering with spatial coordinates and the gene count table.
3. The authors conducted experiments on recent and popular public datasets.

**Weaknesses:**

1. The reviewer is worried that associating/clustering only the spatial coordinates (not the H&E content and spatial coordinates) with Gene might be a problem that has not much usefulness to study. The reviewer understands that this is an important problem but it also always shows up in the first few samples/tutorial when processing a newly harvested ST dataset but it does not provide much further insights other than some visualizations.
2. The reviewer is worried that in order to show a method works well on such ST dataset, we need a more robust or complete or much more samples of ST slides than the ones being compared in the paper. The reviewer does understand that these are the commonly used samples but concerning these samples and their results won’t draw meaningful conclusions.
3. The ablation study is too short to justify each choice made in designing the method.
4. Figure 2, 3, and 5 are not legible enough when printed the paper out and reading in arms length.

**Questions:**

1. The reviewer is not too familiar with all the prior works, how is the training data scope, do the authors train all the spots from several ST together or each training would only contain one single ST’s all spots (which ranges from few hundreds to few thousands)?
2. In the CGCL part, the authors use clusters to create positive and negative labels; there are methods that do not require positive or negative pairs to be defined. Would those methods work here?
3. Could the authors please provide a more detailed ablation study?
4. With all these distribution modeling and guiding bits added, how much does the computation cost or runtime change compared with prior methods?
5. There has been quite a few ST foundation models proposed recently. How do those models perform on such tasks, what is the gap between those zero-shot representations with this single-slide learned representation?


The reviewer is giving a borderline reject recommendation at this stage but happy to modify the score during the discussion phase.

---

> ### Author Response · Authors · 2025-11-21
>
> We are truly grateful for the time you have taken to review our paper and your insightful review. Here we address your comments in the following, where Q denotes the comment of the reviewer and R denotes our response.
>
> > Q1. The reviewer is worried that associating/clustering only the spatial coordinates (not the H&E content and spatial coordinates) with Gene might be a problem that has not much usefulness to study. The reviewer understands that this is an important problem but it also always shows up in the first few samples/tutorial when processing a newly harvested ST dataset but it does not provide much further insights other than some visualizations.
>
> R1. Thanks for your valuable suggestion! **First**, in ST data, gene expression and spatial coordinates constitute the fundamental and most essential primary modalities, whereas the H&E image is merely an auxiliary modality rather than a core component of ST data. Analyzing ST data based solely on gene expression and spatial coordinates is one of the most important tasks in the field. A large number of representative methods, including BayesSpace (NBT’21), CCST (NCS’22), STAGATE (NC’22), PRECAST (NC’23), and DUSTED (AAAI’25), focus exclusively on ST data without using the image modality, and they have been published in top journals and conferences. Therefore, the absence of the image modality does not diminish the scientific value of such research. **Second**, not all ST datasets contain H&E images, and in many cases the image quality is low. Methods such as SpaGCN (NM’21) and stLearn (NC’23) explicitly treat the image modality as optional. Under these circumstances, developing methods that do not rely on the image modality is in fact more broadly applicable and more robust across real-world datasets. **Finally**, while incorporating the H&E image can indeed provide complementary information, it also introduces additional biases and confounding factors, such as non-gene-expression-based noise, histopathological pattern bias, and staining variability. These artifacts may obscure the intrinsic spatial organization encoded by gene expression. Therefore, investigating a cleaner and more focused setting that relies solely on gene expression and spatial coordinates is fully justified and scientifically meaningful.
>
> > Q2. The reviewer is worried that in order to show a method works well on such ST dataset, we need a more robust or complete or much more samples of ST slides than the ones being compared in the paper. The reviewer does understand that these are the commonly used samples but concerning these samples and their results won’t draw meaningful conclusions.
>
> R2. Thanks for your valuable suggestion! We additionally include a new dataset, the mouse olfactory bulb, which was sequenced using the Stereo-seq platform, and contains 19109 spots and 14376 genes across seven tissue layers, i.e., the olfactory nerve layer (ONL), glomerular layer (GL), external plexiform layer (EPL), mitral cell layer (MCL), internal plexiform layer (IPL), granule cell layer (GCL), and rostral migratory stream (RMS). We evaluate the performance of our stUAI, GraphST, and DUSTED on domain identification using this dataset, and report the corresponding Moran’s I scores in the table below.
>
> |  | GraphST | DUSTED | stUAI
> ---|-----|-----|----|
> Moran’s I | 0.59 | 0.28 | 0.67 |
>
> It can be seen that our stUAI yields clustering results with stronger spatial autocorrelation on this dataset. It further validates the effectiveness and generalizability of our method.

---

> ### Author Response · Authors · 2025-11-21
>
> > Q3. The ablation study is too short to justify each choice made in designing the method. Could the authors please provide a more detailed ablation study?
>
> R3. Thanks for your valuable suggestion! Here, we explore more variants to validate the effectiveness and necessity of each module in our method:
> (i) $V_1$: solely adopt the spatial-level graph;
> (ii) $V_2$: solely adopt the feature-level graph;
> (iii) $V_3$: solely adopt cross-view distributional representation alignment;
> (iv) $V_4$: solely adopt cluster-guided intra-view contrastive learning;
> (v) $V_5$: solely adopt ZINB-based reconstruction;
> (vi) $V_6$: replace cluster-guided contrastive learning with standard contrastive learning;
> (vii) $V_7$: replace ZINB-based reconstruction with MSE reconstruction.
>
> | **Dataset** | **Metric** | **V₁** | **V₂** | **V₃** | **V₄** | **V₅** | **V₆** | **V₇** | **stUAI (Ours)** |
> |-------------|------------|--------|--------|--------|--------|--------|--------|--------|------------------|
> | **DLPFC**   | ARI        | 0.61   | 0.30   | 0.52   | 0.66   | 0.60   | 0.81   | 0.80   | **0.84**       |
> |  **(151671)**  | NMI        | 0.69   | 0.22   | 0.66   | 0.72   | 0.68   | 0.74   | 0.75   | **0.78**       |
> | **HBC**     | ARI        | 0.58   | 0.45   | 0.58   | 0.57   | 0.56   | 0.62   | 0.60   | **0.65**       |
> |             | NMI        | 0.63   | 0.52   | 0.62   | 0.63   | 0.62   | 0.65   | 0.64   | **0.69**       |
> | **MAB**     | ARI        | 0.45   | 0.44   | 0.46   | 0.43   | 0.45   | 0.46   | 0.44   | **0.49**       |
> |             | NMI        | 0.68   | 0.63   | 0.67   | 0.66   | 0.67   | 0.66   | 0.67   | **0.71**       |
>
> The comparative results are summarized in the table above. Compared to $V_1$ and $V_2$, we can observe that spatial information plays a more critical role than feature information in domain identification tasks. When comparing $V_3$, $V_4$, $V_5$ with our stUAI, it becomes evident that preserving any single module alone cannot achieve ideal performance, reaffirming that all proposed modules are indispensable. The comparison between $V_6$ and our method demonstrates that the positive samples selected based on clustering confidence provide superior and more stable supervisory signals for contrastive learning. Furthermore, comparing $V_7$ with our approach reveals that MSE fails to adequately capture the sparse and over-dispersion nature of single-cell sequencing data compared to ZINB, consequently leading to performance degradation.
>
> **Please refer to the blue-highlighted content in Table 6, Appendix E in the revised manuscript**.
>
>
> > Q4. The reviewer is not too familiar with all the prior works, how is the training data scope, do the authors train all the spots from several ST together or each training would only contain one single ST’s all spots (which ranges from few hundreds to few thousands)?
>
> R4. Thanks for your valuable problem! Our training is conducted on all spots within a single ST dataset at a time. Each ST dataset consists of a gene expression matrix (rows correspond to spots and columns correspond to genes) together with the spatial coordinates of each spot. We consider three datasets, i.e., DLPFC, HBC, and MAB. The DLPFC dataset contains 12 slices, each of which can be treated as an independent ST dataset, with each slice containing 3460 to 4789 spots and 33538 genes. The HBC and MAB datasets each contain one slice, with 3798 spots and 36601 genes, and 2695 spots and 32285 genes, respectively.
>
>
> > Q5. In the CGCL part, the authors use clusters to create positive and negative labels; there are methods that do not require positive or negative pairs to be defined. Would those methods work here?
>
> R5. Thanks for your valuable suggestion! Here, we replace cluster-guided contrastive learning with standard contrastive learning, which directly treats its own augmented version as positive pairs and all others as negative pairs, eliminating the need for specialized strategies to select positive and negative samples.
>
> | **Dataset** | **Metric** | **Variant** | **stUAI (Ours)** |
> |-------------|------------|--------|------------------|
> | **DLPFC**   | ARI        | 0.81   | **0.84**       |
> |  **(151671)**   | NMI        | 0.74   | **0.78**       |
> | **HBC**     | ARI        | 0.62   | **0.65**       |
> |             | NMI        | 0.65   | **0.69**       |
> | **MAB**     | ARI        | 0.46   | **0.49**       |
> |             | NMI        | 0.66   | **0.71**       |
>
> As can be seen from the table above, when comparing our stUAI with variants using standard contrastive learning, the positive samples selected based on clustering confidence provide more superior and stable supervisory signals for contrastive learning. This enables the learning of more discriminative representations, thereby better serving downstream tasks.
>
> **Please refer to the blue-highlighted content in Table 6, Appendix E in the revised manuscript**.

---

> ### Author Response · Authors · 2025-11-21
>
> > Q6. With all these distribution modeling and guiding bits added, how much does the computation cost or runtime change compared with prior methods?
>
> R6. Thanks for your valuable suggestion! Here we compare the per-iteration running time of two latest methods across two datasets, as shown in the table below.
>
> | **Dataset** | **stLearn** | **DUSTED** | **stUAI (Ours)** |
> |-------------|------------|--------|------------------|
> | **HBC**     | >1s   | 42.25 ms   |    120.63 ms   |
> | **MAB**     | >1s   | 35.10 ms   |    82.17 ms   |
>
> The results show that our method overall maintains a high computational efficiency, positioning itself between two state-of-the-art baselines and significantly lower than stLearn’s runtime. Compared with several baselines, our approach also delivers substantially better performance than all competing methods. This observation further confirms that our method offers superior performance gains while preserving high efficiency.
>
>
> > Q7. There has been quite a few ST foundation models proposed recently. How do those models perform on such tasks?
>
> R7. Thanks for your valuable suggestion! Here, we select two ST foundation models (one highly authoritative model Geneformer (Nature'23) and one latest model Nicheformer (NM'25)) to validate the effectiveness of our approach. The comparative results are shown in the table below.
>
> | **Dataset** | **Metric** | **Geneformer (Nature'23)** | **Nicheformer (NM'25)** |   **stUAI (Ours)** |
> |-------------|------------|--------|------------------|----------|
> | **DLPFC**   | ACC       | 0.6404   |  0.5712   |   **0.6921**    |
> | **(151673)**    | F1        | 0.5770   |  0.5319   |    **0.6024**   |
> | **DLPFC**   | ACC       | 0.6286   |  0.5942   |   **0.6547**    |
> | **(151674)**   | F1        | 0.5464   |  0.5300   |  0.5323     |
> | **DLPFC**   | ACC       | 0.6373   |  0.5588   |  **0.7185**     |
> | **(151675)**    | NMI       | 0.5677   |  0.5192   |  **0.6437**     |
> | **DLPFC**   | ACC       | 0.6157   |  0.5648   |  0.6042     |
> | **(151676)**    | F1        | 0.5513   |  0.5141   |   0.5151    |
> |     | **Average ACC**       | 0.6305   |  0.5723   |   **0.6674**    |
> |     | **Average F1**        | 0.5606   |  0.5238   |   **0.5734**    |
>
> As can be seen from the results, our stUAI achieve optimal performance in most cases across the four datasets. It also surpasses two competitive methods in terms of average performance, which fully demonstrates the superiority of our proposed stUAI over existing approaches.
>
> [1] Theodoris C V, Xiao L, Chopra A, et al. Transfer learning enables predictions in network biology[J]. Nature, 2023, 618(7965): 616-624.
>
> [2] Tejada-Lapuerta A, Schaar A C, Gutgesell R, et al. Nicheformer: a foundation model for single-cell and spatial omics[J]. Nature Methods, 2025: 1-14.
>
> In light of these responses, we hope we have addressed your concerns, and hope you could consider raising your score. If there are any additional notable points of concern that we have not yet addressed, please do not hesitate to share them, and we will promptly attend to those points.

---

> ### Comment · Reviewer_bht3 · 2025-11-25
> **Ack**
>
> The reviewer appreciates the detailed rebuttal provided by the authors. Most of my questions are being addressed. I am happy to raise my rating to a 6. Good luck.

---

> > ### Author Response · Authors · 2025-11-26
> >
> > Dear Reviewer bht3,
> >
> > Thanks for your feedback and increasing the rating! We are pleased to know that our responses have addressed your concerns. We will add the rebuttal contents to the main paper in the final version following your valuable suggestions.
> >
> > Best regards,
> >
> > the Authors

---

### Author Response · Authors · 2025-11-30

Dear Area Chairs,

We would like to express our sincere gratitude for stepping in to oversee our submission following the recent technical issues on OpenReview, as well as for your valuable time and effort dedicated to the review process. To facilitate your efficient evaluation, we have prepared a concise summary of the current rebuttal and reviewer interaction status:

### **Overall Interaction Overview**

Our manuscript received initial reviews from four reviewers. We have submitted a detailed rebuttal supplemented with additional experiments, comprehensively addressing all raised concerns: Reviewer bht3 has raised the rating from 4 to 6, Reviewer XHFa’s remaining request (only one) has been fully resolved, and the other two reviewers (Rfbg and 2U3y) have not yet responded but their core questions have been adequately addressed.

### **Resolved Reviewer Feedback**

**Reviewer bht3 (Rating: 4 → Updated to 6)**

The reviewer has already confirmed “**his questions have been addressed and is happy to raise the rating to 6**,” indicating full acceptance of our responses.

**Reviewer XHFa (Rating: 4, only one remaining request, which has been fully addressed)**

The reviewer acknowledged that “**most concerns have been addressed**” and **only requested** supplementary average results across all 12 slices of the DLPFC dataset to verify the method’s robustness. We have fulfilled this request by adding the required experiments in the revised manuscript’s appendix (Tables 7, 8, and Figure 10 in Appendix E), which comprehensively demonstrate the consistency and generalization of our method across different slices.

### **Core Concerns of Non-Responding Reviewers & Our Solutions**

**Reviewer Rfbg (Rating: 4, has not yet responded)**

The reviewer raised concerns regarding dataset diversity and fusion strategy comparisons, all of which have been addressed with supplementary analyses. We have fully resolved these points by:
- adding a new Stereo-seq mouse olfactory bulb dataset (with 19109 spots and 7 tissue layers) to robustly verify generalization;
- conducting comparisons between our uncertainty-aware fusion and attention-based fusion, clearly demonstrating our method’s superiority across key metrics.

**Reviewer 2U3y (Rating: 4, has not yet responded)**

The reviewer’s feedback centered on clarificatory and supplementary requests (**rather than fundamental flaws**), including novelty clarification, uncertainty details, and extended validations. We have comprehensively addressed each point through:
- emphasizing that **we are the first** to tailor probabilistic embedding to SRT for spot-level distributional representations, plus three SRT-specific innovations forming a unified self-supervised framework;
- explicitly defining uncertainty as data-inherent noise/graph biases with concrete utility in fusion weighting;
- supplementing comparisons between OT and KL/JS divergences to validate our method selection;
- proposing actionable strategies for unimodal data adaptation;
- integrating the previously missing reference.

In summary, we have thoroughly addressed both the initial comments and the existing follow-up queries. We respectfully request your consideration of these updates when making your final decision.

Thank you again for your hard work.

Sincerely,

The Authors

---

### Meta-Review · Area_Chair_StVs · 2026-01-07

**Summary:**

This paper poproses stUAI for ST data. stUAI learns spatial and genomic embeddings using graph-based encoders and use distributional representation at spot-level to quantify uncertainty, which is sometimes overlooked in the existing literature. Sparsity and spatial continuity are also considered in the model. Experimental results support the model's effectiveness in spatial clustering and some other tasks. All four reviewers found the paper easy to follow and the uncertainty quantification novel and interesting. However, there are some major concerns, where some were addressed during the rebuttal, some remain. In my opinion, there are a few remaining key issues that prevent me to recommend to accept in the current form, briefly summarized below (see a more detailed list in the next block):

1. For a single slide, the experiments, especially the clustering experiments seem comprehensive. However, the experiments focus on 2 slides only, with one more MAB slide during the rebuttal. This is insufficient in my opinion given the well known heterogeneity of ST data. For example, different tissue types display very different spatial genomic pattern. Even for the same dataset, say DFPLC, there are 12 slides but the paper only present results from slide 151507. I strongly recommend the authors to include results from other slides in the appendix at least.
2. I agree with reviewers that the uncertainty is a big plus of the paper. However, the utility of the uncertainty is not as deeply discussed as for clustering tasks. The authors mitigated this question from some reviewers by adding Appendix H, which gives some high level ideas and suggestions. I appreciate this addition but I do think doing some concrete downstream analysis with concrete results will make the paper much stronger.
3. Comparison with ST foundation model is expected, as raised by one reviewer. Even with the new results on Nichformer during the rebuttal, it's far from comprehensive. As listed in the block below, there are quite a few existing foundation models even if we exclude those from unpublished preprints (all models I listed below are from published work).

Addressing these remaining concerns in the next version will make the paper stronger and I do believe there will be a favorable decision given the strengths of this paper.

**Reviewer Concerns:**

Reviewer bht3
1. Clustering excludes H&E (addressed by comprehensive explanation)
2. Limited scope of real data (addressed by additional experiments on a mouse olfactory bulb slice from Stereo-seq platform)
3. Ablation study is too short (addressed by more ablation studies)
4. Comparison with standard contrastive learning (addressed by additional experiments)
5. Missing computational cost discussion (addressed by reporting runtime)
6. Comparison with existing ST foundation models (addressed by adding results from Geneformer and Nichformer)
Note from the AC: Geneformer is not generally considered as a ST foundation model. Instead, there are some other existing ST foundation models such as stFormer, SToFM, STPath, OmniCLIP, etc.

Reviewer XHFa
1. Time complexity (addressed by providing runtime)
2. Justification of the architecture (addressed by detailed explanation)
3. Motivation is not strong enough (addressed by detailed explanation)
4. Sensitivity analysis (addressed by new experiments)
5. Unusual pseudotime trajectory (authors admit that there was some error, and the figure was fixed during the rebuttal)
6. Experiments focus on a single slice (addressed by adding more results on more slides)

Reviewer Rfbg
1. More dataset (partially addressed by adding one new slide)
2. Comparison with attention-based fusion (addressed by the requested experiment)
3. Interpretation of uncertainty (partially addressed by clarifying the notion of uncertainty in the proposed model framework)

Reviewer 2U3y
1. Limited novelty (partially addressed by clarifying the contribution and novelty)
2. Source of uncertainty (adressed by further discussion)
3. Utility of uncertainty (partially addressed by discussing some potential utilities without concrete results)
4. Benefit to unimodal data (addressed by detailed discussion)

**Reviewer Scores:**

Reviewer bht3: 4 --> 6

Reviewer XHFa: 4 --> 5

Reviewer Rfbg: 4 --> 4

Reviewer 2U3y: 4 --> 4

---

### Decision · Program_Chairs · 2026-01-26

Reject